# Bottom-up precise synthesis of stable platinum dimers on graphene

Huan Yan[1], Yue Lin[2], Hong Wu[1], Wenhua Zhang[3], Zhihu Sun[4], Hao Cheng[4], Wei Liu[4], Chunlei Wang[1], Junjie Li[1], Xiaohui Huang[1], Tao Yao [4], Jinlong Yang[1,2], Shiqiang Wei[4] & Junling Lu [1,2,3,5]

Supported metal clusters containing only a few atoms are of great interest. Progress has been made in synthesis of metal single-atom catalysts. However, precise synthesis of metal dimers on high-surface area support remains a grand challenge. Here, we show that $Pt_2$ dimers can be fabricated with a bottom–up approach on graphene using atomic layer deposition, through proper nucleation sites creation, $Pt_1$ single-atom deposition and attaching a secondary Pt atom selectively on the preliminary one. Scanning transmission electron microscopy, x-ray absorption spectroscopy, and theoretical calculations suggest that the $Pt_2$ dimers are likely in the oxidized form of $Pt_2O_x$. In hydrolytic dehydrogenation of ammonia borane, $Pt_2$ dimers exhibit a high specific rate of 2800 $mol_{H2}$ $mol_{Pt}^{-1}$ $min^{-1}$ at room temperature, ~17- and 45-fold higher than graphene supported Pt single atoms and nanoparticles, respectively. These findings open an avenue to bottom–up fabrication of supported atomically precise ultrafine metal clusters for practical applications.

[1] Department of Chemical Physics, University of Science and Technology of China, Hefei, Anhui 230026, China. [2] Hefei National Laboratory for Physical Sciences at the Microscale, University of Science and Technology of China, Hefei, Anhui 230026, China. [3] CAS Key Laboratory of Materials for Energy Conversion, University of Science and Technology of China, Hefei, Anhui 230026, China. [4] National Synchrotron Radiation Laboratory, University of Science and Technology of China, Hefei, Anhui 230029, China. [5] Collaborative Innovation Center of Chemistry for Energy Materials (iChEM), University of Science and Technology of China, Hefei 230026, China. Huan Yan and Yue Lin contributed equally to this work. Correspondence and requests for materials should be addressed to S.W. (email: sqwei@ustc.edu.cn) or to J.L. (email: junling@ustc.edu.cn)

Supported metal catalysts are among the most important categories of heterogeneous catalysts in many reactions including chemical upgrading, automobile exhaust treatment, Fischer-Tropsch synthesis, biomass conversions, and many other processes[1–7]. Decreasing metal particle size is desirable for improving metal utilization, since catalytic reactions take place on the surface of metal nanoparticles (NPs). When a metal cluster contains only a few metal atoms, it could have a discrete energy band structure, tightly correlated with the number of metal atoms. Changing one atom in the ultrafine cluster might largely alter the electronic structure and drastically change its catalytic properties. Such atom-dependent catalytic behaviors have been successfully demonstrated by the model catalysts of mass-selected metal clusters, which were fabricated by soft landing of mass-selected ions from their physical vapor under ultrahigh vacuum conditions[8–14]. However, such complicated approach is only limited to model catalyst studies and is not applicable to high-surface area supports for practical applications.

Recently, synthesis of supported metal single-atom catalysts (SACs) has been extensively explored and a number of successful examples have been demonstrated[15–23]. Nonetheless, synthesis of atomically precise ultrafine metal clusters, such as dimers, on high-surface area supports, remains a grand challenge. The decisive limitation is the lack of precise control over the aggregation process, which often causes metal NPs formation with a broad size distribution. Protecting metal clusters with a strong ligand can certainly inhibit metal aggregation to a large extent, such as in the case of thiolate-protected Au magic clusters[24]. However, these strong protective ligands typically poison the metal clusters, and decrease their catalytic activities considerably[25–30]. Alternatively, Gates et al. demonstrated that precisely defined iridium and rhodium clusters were achieved by grafting the corresponding carbonyl complexes with a specific number of metal atoms onto oxide supports[31–33]. But the success is limited. As a consequence, a general bottom–up approach to synthesize atomically precise metal clusters on high-surface area supports is still missing. Atomic layer deposition (ALD) relies on two sequential self-limiting surface reactions at the molecular level, which are separated by inert gas purging[34–36]. This unique character makes ALD possible to bottom–up construct catalytic materials on a high-surface area substrate uniformly and precisely[37–39].

Here, we show that Pt$_2$ dimers can be bottom–up fabricated on a graphene support by depositing Pt on phenol-related oxygen anchor sites atom-by-atom in a sequential manner using Pt ALD. The dominant presence of isolated Pt$_1$ single atoms and Pt$_2$ dimers in the corresponding samples were confirmed by both aberration-corrected high-angle annular dark-field scanning transmission electron microscopy (HAADF-STEM) and X-ray absorption fine structure spectroscopy (XAFS). Their structures were determined through a combination of density function theory (DFT) calculations and XAFS spectra simulations. In hydrolysis of ammonia borane (AB) for hydrogen generation, graphene supported Pt$_2$ dimers (Pt$_2$/graphene) exhibited a striking activity, which is ~17- and 45-fold higher than that of graphene supported Pt$_1$ single atoms and Pt NPs, respectively. Compared to Pt$_1$ single atoms and Pt NPs, the decreased adsorption energies of both AB and H$_2$ molecules on Pt$_2$ dimers are likely the major reason for the high activity. More importantly, the Pt$_2$ dimers were stable under the current reaction condition and in the inert environment at below 300 °C.

## Results

### Synthesis and morphology of Pt$_1$/graphene and Pt$_2$/graphene.

Based on our recent strategy[22], the nucleation sites of isolated phenols or phenol–carbonyl pairs suggested by Shenoyl et al[40]. for Pt ALD were first created on pristine graphene nanosheets through acid oxidation followed by high-temperature thermal reduction, as illustrated in the schematic model in Fig. 1. The reduced graphene oxide support was defect-rich multilayered graphene films with a thickness of about a few nanometers (Supplementary Fig. 1). It had a surface area of 570 m$^2$/g. Next, Pt ALD was performed on the graphene support by alternately exposing trimethyl(methylcyclopentadienyl)-platinum(IV) (MeCpPtMe$_3$) and molecular O$_2$ at 250 °C. The self-limiting surface reactions between MeCpPtMe$_3$ and the support ensure nucleation of one MeCpPtMe$_3$ molecule on one phenol-related nucleation site during the saturated MeCpPtMe$_3$ exposure (Supplementary Fig. 2). It should be noted that Pt nucleation on graphene defect sites, such as edges and line defects, was inhibited at 250 °C, although it is possible at 300 °C (Supplementary Table 1)[41]. A similar temperature effect on inhibiting metal ALD on oxides was also observed by Elam et al[42]. Next, the ligands were removed through a combustion reaction during the O$_2$ exposure step[43–45] and Pt$_1$ single atoms are formed (denoted as Pt$_1$/graphene).

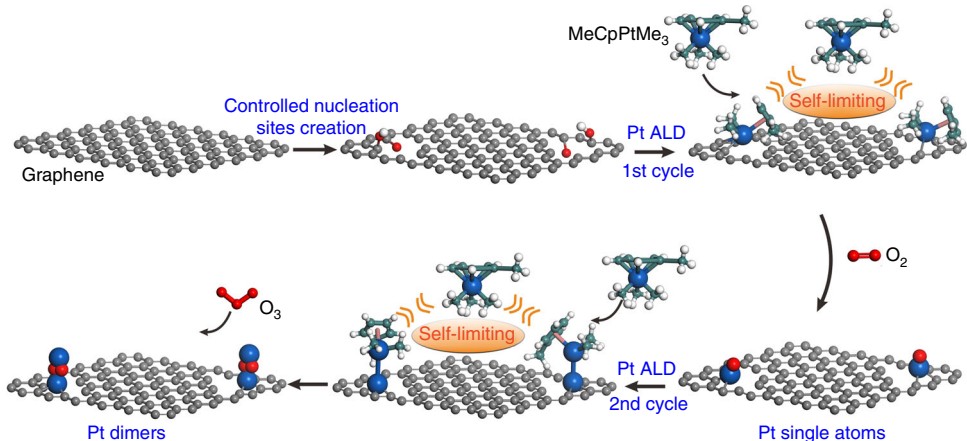

**Fig. 1** Schematic illustration of bottom–up synthesis of dimeric Pt$_2$/graphene catalysts. Controlled creation of isolated anchor sites on pristine graphene; one cycle of Pt ALD on the anchor sites for Pt single atoms formation by alternately exposing MeCpPtMe$_3$ and molecular O$_2$ at 250 °C; second cycle of Pt ALD on the Pt$_1$/Graphene to selectively deposit the secondary Pt atoms on the preliminary ones for Pt$_2$ dimers formation at 150 °C. The balls in cyan, white, red, and blue represent carbon, hydrogen, oxygen, and platinum while the ball in gray represents carbon atoms in the graphene support

Next, the formed $Pt_1$ single atoms was further utilized as nucleation sites for anchoring the secondary $MeCpPtMe_3$ molecule in the following cycle. Again, the steric hindrance between $MeCpPtMe_3$ molecules restricts chemisorbing one $MeCpPtMe_3$ molecule only on one isolated $Pt_1$ atom. However, we noticed that a considerable amount of Pt NPs were formed after two successive cycles of Pt ALD at 250 °C (denoted as 2cPt/graphene, Supplementary Fig. 3). Therefore, the deposition temperature was decreased to 150 °C for the second ALD cycle to avoid any metal aggregation. Meanwhile, ozone ($O_3$), a stronger oxidizing regent was utilized to remove the ligand efficiently to form $Pt_2$ dimers ($Pt_2$/graphene) (Fig. 1)[46]. The formed $Pt_1$ single atoms and $Pt_2$ dimers are expected to be in the oxidized forms, since they were exposed to $O_2$ and $O_3$ during synthesis, respectively.

Aberration-corrected HAADF-STEM measurements were carried out to investigate the morphologies of the single-atom $Pt_1$/graphene and dimeric $Pt_2$/graphene catalysts. Compared to the naked graphene (Supplementary Fig. 1), HAADF-STEM images illustrated that one cycle of Pt ALD on the graphene support at 250 °C resulted in a formation of atomically dispersed $Pt_1$ atoms without presence of any visible clusters or NPs (Fig. 2a–c, and Supplementary Fig. 4). These $Pt_1$ single atoms were well isolated from each other with a distance >2 nm in average, which is significantly larger than the effective diameter of the $MeCpPtMe_3$ molecule of ~0.96 nm[47], confirming the steric effect during synthesis. Similar to our recent study of $Pd_1$ single-atom growth on graphene[22], we found that complete removal of other oxygen-contained functional groups, such as carboxyl groups, from graphene by carefully tuning the reduction temperature and time, is the key to eliminate any Pt clusters or NPs formation (Supplementary Fig. 5). These findings suggest that metal atoms anchored on carboxyl groups have a general weak interaction with the graphene support, thus aggregate aggressively to larger metal NPs under the ALD conditions, in line with literature[48].

After performing another cycle of Pt ALD on $Pt_1$/graphene at 150 °C ($Pt_2$/graphene), $Pt_2$ dimers were dominantly formed along with a certain number of $Pt_1$ single atoms (Fig. 2d–f, and Supplementary Fig. 6), where neither Pt clusters nor NPs were observed. Very interestingly, we noticed that $Pt_2$ dimers frequently rotated by specified angles of 30, 60, and 90° under the electron beam during STEM measurements and then split into two isolated $Pt_1$ atoms (see more details in Supplementary Figs. 7–9). Such characteristic rotations might be related with the geometry of the graphene support and the size of carbon defect by considering the aforementioned $Pt_2$ dimer structure (Supplementary Fig. 10). This observation provides strong evidence of the presence of $Pt_2$ dimers rather than the projection coincidence of two isolated $Pt_1$ atoms at different Z positions. Statistical analysis of more than 80 pairs of $Pt_2$ dimers showed a Pt–Pt distance of 0.30 ± 0.02 nm for $Pt_2$ dimers (Fig. 2g), which is significantly longer than the Pt–Pt bond in Pt bulk. This indicates that the $Pt_2$ dimers are in the oxidized form as expected.

To step-wise elucidate the selective deposition of secondary Pt atom onto the preliminary ones for the formation of $Pt_2$ dimers in the second ALD cycle, a set of control experiments were further performed using inductively coupled plasma-atomic emission spectroscopy (ICP-AES). First, the influence of the Pt precursor ligand on the second Pt ALD cycle was examined. In this case, half cycle of Pt ALD was executed on the graphene support by performing the $MeCpPtMe_3$ pulse step only (denoted as MeCpPtMe/graphene) at 250 °C. After that, the ALD reactor was cooled to near room temperature and half amount of the MeCpPtMe/graphene sample was taken out of reactor for ICP-AES analysis; while the rest of the sample was put back to the

ALD reactor quickly to perform the second cycle (Pt-MeCpPtMe/graphene) at 150 °C. In nine independent trials, the ICP-AES results showed that the ratio of the Pt loadings of Pt-MeCpPtMe/graphene to those of the corresponding MeCpPtMe/graphene were all very close to one (Fig. 2h). Therefore, there was no additional Pt deposited on MeCpPtMe/Graphene, reflecting the saturated self-limiting reaction character of ALD[34–36]. Second, we found that exposure of pristine graphene to $O_2$ at 250 °C did not cause any detectable Pt deposition either (Supplementary Table 1). This is very important to ensure that the oxygen pulse at 250 °C in the first ALD cycle did not create any additional nucleation sites. Taken together, the $Pt_1$ single atoms formed on graphene, confirmed by HAADF-STEM in Fig. 2a–c, are the only nucleation sites for the following ALD cycle. It is worthy to note that $Pt_1$ single atoms well isolated from each other could be crucial to make all the $Pt_1$ single atoms accessible for chemisorbing the second $MeCpPtMe_3$ molecule in the second ALD cycle without steric hindrance.

During the second ALD cycle, one $MeCpPtMe_3$ molecule anchors on one $Pt_1$ atom in the $Pt_1$/graphene sample due to the steric hindrance effect, which doubles the Pt loading. This was confirmed by the ratios of two for the Pt loadings of $Pt_2$/Graphene to $Pt_1$/graphene in nine independent trials (Fig. 2i and Supplementary Table 2), hence providing strong evidence of the formation of $Pt_2$ dimers. The $Pt_1$ single atoms observed in Fig. 2e and Supplementary Figs. 6–9 were likely formed by uncoupling of $Pt_2$ dimers under the high flux electron beam during STEM measurements[49]. On the other hand, once Pt NPs were formed during ALD, the ratio of Pt loading of the two-ALD cycle sample to the one-cycle sample was apparently off from the stoichiometric value of two (Supplementary Table 2).

Performing an additional cycle on $Pt_2$/graphene to form $Pt_3$ trimers might be possible. However, we noticed that the $O_3$ in the second cycle can create additional nucleation sites on graphene. As a consequence, selective deposition was not achieved for the third cycle and resulted in a mixture of $Pt_1$, $Pt_2$ and $Pt_3$. Therefore, we mainly focused on the $Pt_2$ dimers in this work.

**XAFS characterization and DFT calculations.** Figure 3a shows the X-ray absorption near-edge structure (XANES) spectra of MeCpPtMe/graphene, $Pt_1$/graphene, and $Pt_2$/graphene at the Pt $L_3$-edge, along with Pt foil, $PtO_2$, and $MeCpPtMe_3$ as references. Evidently, the XANES white line peaks of these three samples (11,567 eV) located at between Pt foil and $PtO_2$, indicating that the Pt in MeCpPtMe/graphene, $Pt_1$ single atoms, and $Pt_2$ dimers were all in a similar oxidation state between $Pt^0$ and $Pt^{4+}$. The $MeCpPtMe_3$ reference sample exhibits two well-resolved peaks at 1.62 and 1.99 Å in the Fourier transformed (FT) $k^3\chi(k)$ curve in the real-space ($R$-space) (Fig. 3b), assigned to the shorter Pt–C bonds (1.99–2.14 Å) in the three Pt–Me groups and longer Pt–C bonds (2.26–2.36 Å) in the Pt–MeCp group, respectively (Supplementary Fig. 11a)[47,50]. Apparently, the two split peaks in the MeCpPtMe/graphene curve suggests that the MeCp group stayed on Pt in this sample. This observation is in line with both the experimental result of the formation of $MeCpPtMe_2$ surface species on oxides[43,51] and theoretical calculations where $MeCpPtMe_3$ on epoxydated and hydroxylated graphene surfaces liberates either one or two methyl groups depending on the available surface groups[52].

The first shell FT peak in the $Pt_1$/graphene spectrum had a higher intensity and slightly shifted to 1.65 Å, while the peak at 1.99 Å disappeared. Clearly, the MeCp ligand was combusted off after the $O_2$ exposure step at 250 °C. The first shell peak is assigned to Pt–O and/or Pt–C coordinations. Similar to MeCpPtMe/graphene, a very weak peak at 2.4 Å was visible in

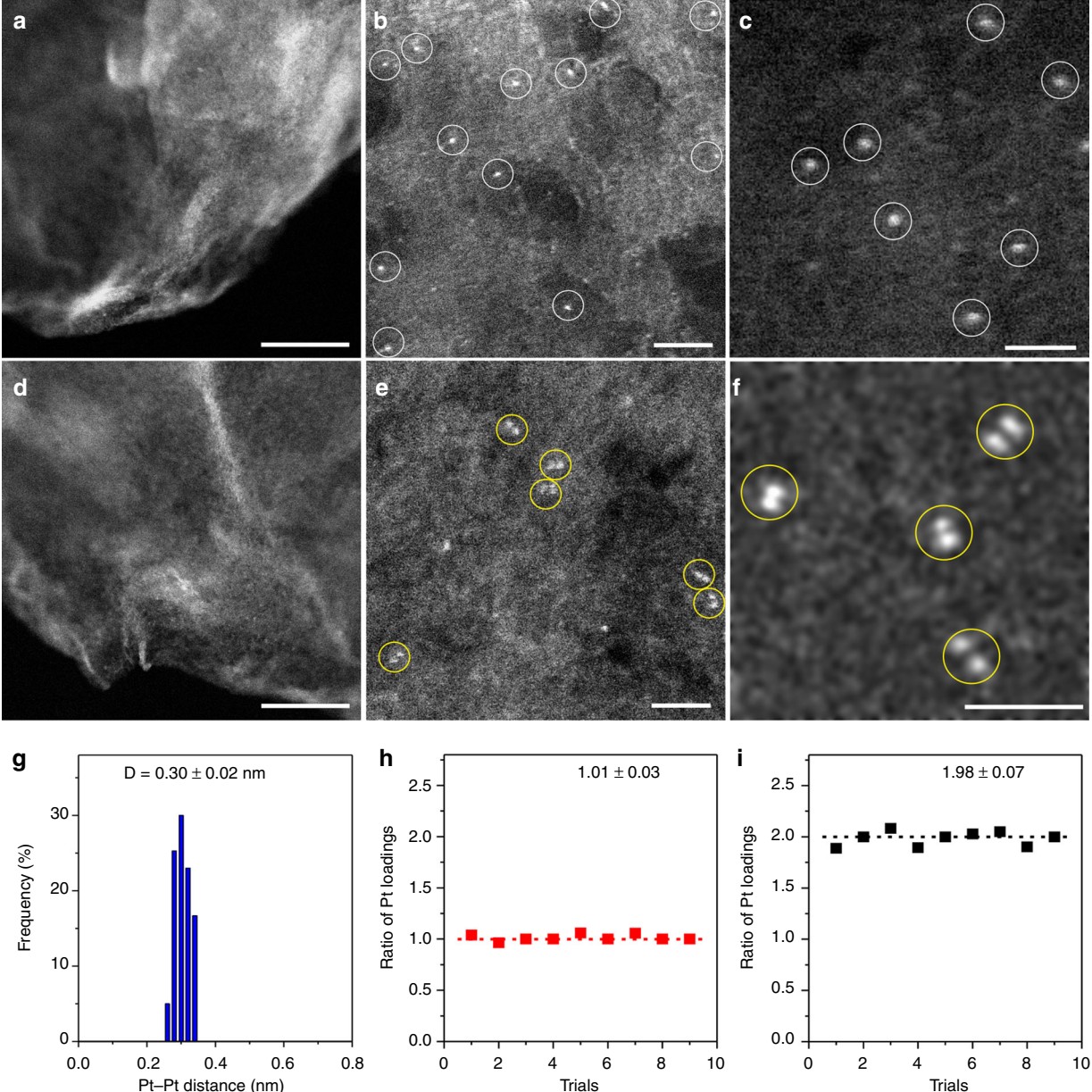

**Fig. 2** Morphology of the single-atom $Pt_1$/graphene and dimeric $Pt_2$/graphene catalysts. Aberration-corrected HAADF-STEM images of $Pt_1$/graphene (**a–c**) and dimeric $Pt_2$/graphene (**d**, **e**). Scale bars, 20 nm (**a**, **d**), 2 nm (**b**, **e**), and 1 nm (**c**, **f**). Pt single atoms in **b** and **c** and dimers in **e** and **f** are highlighted by white and yellow circles, respectively. **g** Statistical Pt–Pt distance in the observed $Pt_2$ dimers. **h** The ratio of Pt loading of in Pt-MeCpPtMe/graphene to that in MeCpPtMe/graphene, and **i** the ratio of Pt loading in $Pt_2$/graphene to that in $Pt_1$/graphene in nine independent trials determined by ICP-AES

the spectrum of $Pt_1$/graphene. However, this peak is significantly different from the Pt–Pt coordination peak (at ~2.62 Å) in Pt foil, thus assigned to the second nearest C/O neighbors of Pt. This suggests the absence of Pt NPs in $Pt_1$/graphene, consistent with our STEM observation (Fig. 2a–c and Supplementary Fig. 4). The dimeric $Pt_2$/graphene sample showed a similar FT curve with $Pt_1$/graphene, implying a similar local C/O coordinations in these two samples. In the $Pt_2$/graphene spectrum, there was no discernible peak for the Pt–Pt coordination, suggesting the $Pt_2$ dimers are in the oxidized form after the ozone exposure step at 150 °C.

Considering the difficulties in discriminating the C/O neighbors by EXAFS fittings, we resorted to the combination of DFT calculations with EXAFS simulations to determine the optimized structures of these three samples. Here, a graphene

support containing a carbon vacancy along with either isolated phenol group or phenol–carbonyl pairs[40] was employed as the reduced graphene oxide surface. The structural models optimized by DFT calculations were further examined by EXAFS simulations.

Regarding the previous work[52], the structures of $MeCpPtMe_2$ and MeCpPtMe were both considered for the MeCpPtMe/graphene sample (Supplementary Fig. 11). Compared to the $MeCpPtMe_3$ molecule, the five Pt–C bonds in the Pt–MeCp group in these two structures both changed significantly. According to the EXAFS simulations for these two structures, MeCpPtMe bonded to the graphene support through two interfacial O atoms might be the promising structure for the MeCpPtMe/graphene sample (Fig. 3c and Supplementary Figs. 11b and 12). Compared to the spectrum of $MeCpPtMe_3$,

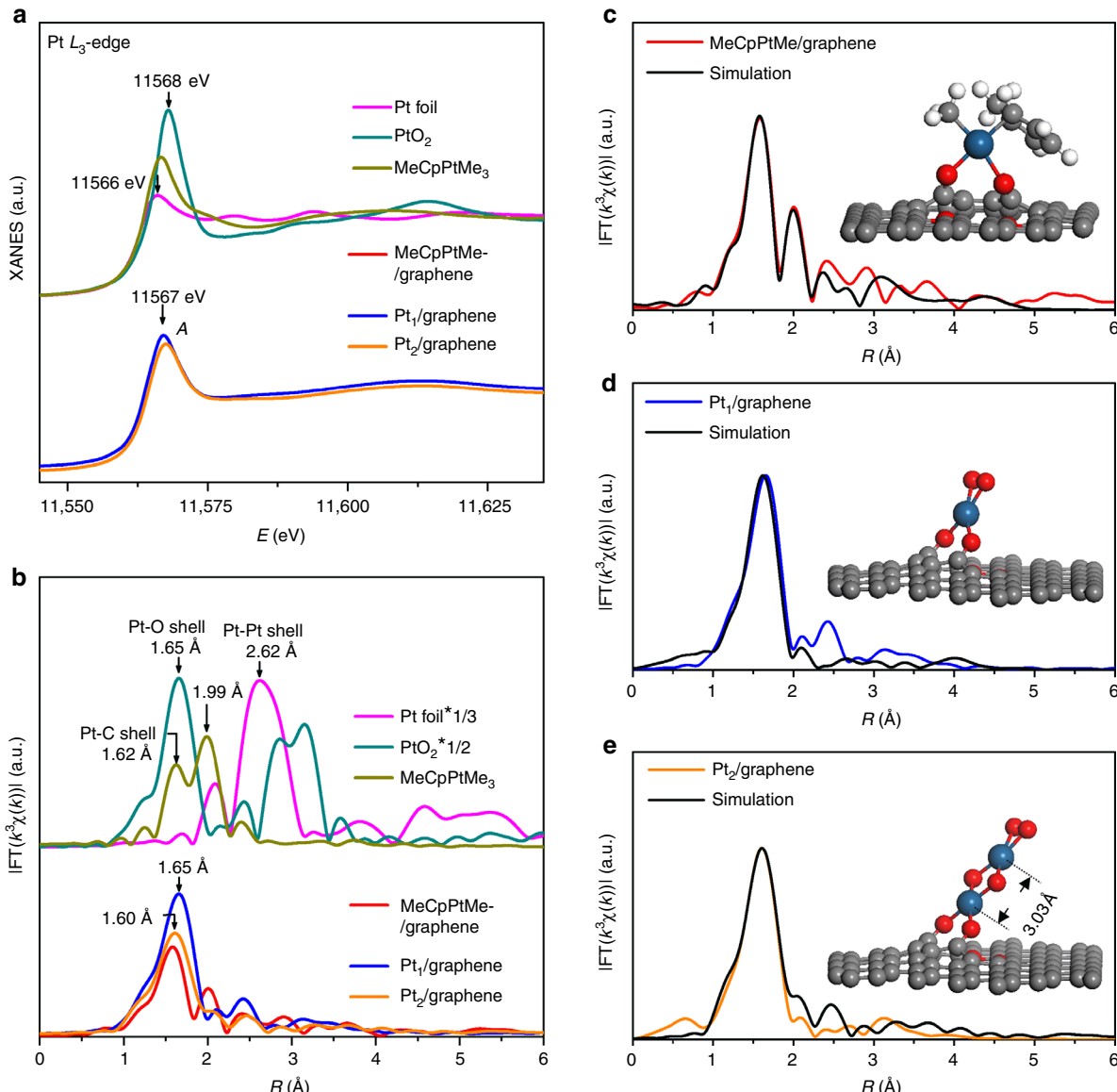

**Fig. 3** XAFS structural characterization and spectra simulations. **a** The XANE spectra and **b** the $K^2$-weighted Fourier transform spectra of MeCpPtMe/graphene, $Pt_1$/graphene, and $Pt_2$/graphene at the Pt $L_3$-edge. The reference samples of Pt foil, $PtO_2$, and $MeCpPtMe_3$ are also shown for comparison. Comparison of the EXAFS simulations based on the corresponding DFT calculated structural models (insets) with the experimental EXAFS spectra of MeCpPtMe/graphene (**c**), $Pt_1$/graphene (**d**) and $Pt_2$/graphene (**e**). The ball in gray, white, red, and dark blue represent carbon, hydrogen, oxygen, and platinum, respectively

the remarkably attenuated peak at 1.99 Å in the MeCpPtMe/graphene spectrum is due to the considerable distortion of the MeCp group.

When oxygen combusts off the ligand, additional oxygen chemisorbs on the $Pt_1$ atom in the $Pt_1$/graphene sample[53]. Indeed, $Pt_1$ atom with one chemisorbed $O_2$ molecule at the terminal position (the Pt–O bond distance: 2.00 Å) and two O atoms at the interface (the Pt–O bond distance: 2.02 Å) produces an EXAFS spectrum in good agreement with the experimental result (Fig. 3d and Supplementary Fig. 12). On the contrary, the $Pt_1$ atom with one O and one C atom at the interface generates split FT peaks in the first shell, in contrast with the experimental results (Supplementary Fig. 13). Nonetheless, this structure might not be completely ruled out.

During the second Pt ALD cycle, a secondary Pt atom anchors on the preliminary one and then becomes oxidized during the $O_3$

exposure step. Taking this information into account, a $Pt_2O_6$ chain structure with O atoms alternating between the terminal and bridge positions was constructed (inset in Fig. 3e). After optimization, our calculations show that the Pt–Pt bond distance in the $Pt_2O_6$ chain is 3.03 Å (the inset of Fig. 3e), consistent with the experimental results very well (Fig. 2g). The lengths of the Pt–O bonds in the $Pt_2O_6$ chain are very close to each other, ~1.93–2.03 Å. Moreover, XAFS spectrum simulation for this $Pt_2O_6$ chain structure also agrees very well with the experimental result (Fig. 3e and Supplementary Fig. 12). This chain structure is found to be similar to the suggested structure models for $Pt_xO_y$ ($x$ = 1–3) clusters by Schneider et al. previously[54]. Interestingly, we also noticed that the tilted angle of the $Pt_2O_6$ chain could vary largely from ~8 to ~50°, depending on both the size of carbon vacancies and the configurations of two interfacial O atoms (Supplementary Fig. 14). The largely varied angles tilted from the

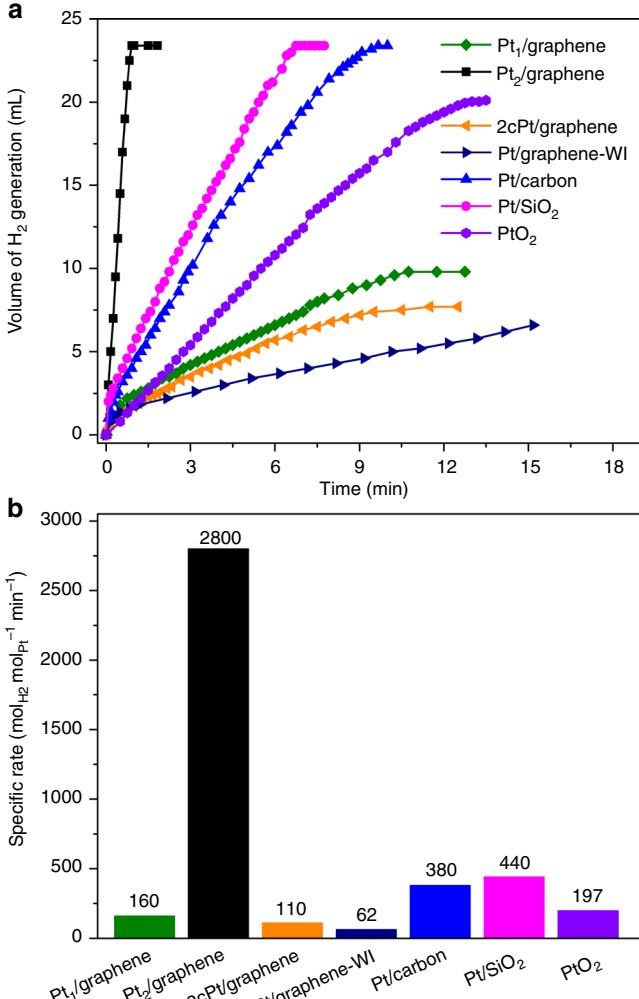

**Fig. 4** Catalytic activities of various Pt catalysts in AB hydrolysis. **a** Plots of time vs normalized volume of hydrogen gas generated from the AB hydrolysis reaction over the single-atom $Pt_1$/graphene and dimeric $Pt_2$/graphene catalysts. Pt NPs samples of 2cPt/graphene, Pt/graphene-WI, Pt/carbon, Pt/SiO$_2$, and commercial PtO$_2$ powder were also evaluated as a comparison. **b** The specific rates over these samples based on the mole of Pt

graphene support, explain well the different Pt–Pt bond distances in the Pt$_2$ dimers observed by STEM (Fig. 2e–g). Again, the structure model of Pt$_2$ dimers with one O and one C atom at the interface might not be completely ruled out.

**Catalytic activity**. AB with satisfactory air stability and remarkably high hydrogen content of 19.6 wt%, has been regarded as a promising hydrogen storage media for portable applications[55]. Here, hydrolysis of AB for hydrogen production was utilized as a probe reaction to investigate the catalytic properties of Pt$_1$ single atoms and Pt$_2$ dimers. According to literature, this reaction is depicted as the following Eq. (1):[56]

$$NH_3BH_3 + 2H_2O \rightarrow NH_4^+ + BO_2^- + 3H_2(g) \quad (1)$$

As shown in Fig. 4a, the single-atom Pt$_1$/graphene catalyst generated 9.8 mL H$_2$ gas only in 10.8 min, which is ~42% of the theoretical volume of 23.4 mL according to the Eq. (1). In sharp contrast, the dimeric Pt$_2$/graphene catalyst released 23.4 mL H$_2$ vigorously in only 0.9 min, indicating the depletion of AB.

The activity of 2cPt/graphene, synthesized by two successive cycles of Pt ALD on graphene at 250 °C, was rather close to Pt$_1$/graphene, by generating 7.5 mL H$_2$ in 10.5 min. Obviously, Pt$_2$/graphene and 2cPt/graphene were distinctly different in structure. As a comparison, the activities of the Pt NP catalysts of Pt/graphene-WI, Pt/carbon, and Pt/SiO$_2$, as well as the commercial PtO$_2$ were tested. The Pt/graphene-WI catalyst with a Pt particle size of $1.8 \pm 0.5$ nm (Supplementary Fig. 15 and Table 3), showed a very poor activity of 6.6 mL H$_2$ release in 15.2 min. The commercial Pt/carbon catalyst with a Pt particle size of $2.3 \pm 0.7$ nm (Supplementary Fig. 16 and Table 3) was considerably better, generating 23.4 mL H$_2$ in 9.7 min. The Pt/SiO$_2$ ALD catalyst with a Pt particle size of $1.9 \pm 0.3$ nm (Supplementary Fig. 17 and Table 3) was also very active, releasing 23.4 mL H$_2$ in 6.8 min. The commercial PtO$_2$ powder (Supplementary Fig. 18) generated ~ 21 mL H$_2$ in 15 min. In this case, a reduction of PtO$_2$ into Pt occurred during the reaction, in line with literature[56].

The specific rates of these samples were calculated based on the Pt contents. The rates were 160 and 110 mol$_{H2}$ Mol$_{Pt}$$^{-1}$ min$^{-1}$, for Pt$_1$/graphene and 2cPt/graphene, respectively (Fig. 4b and Supplementary Fig. 19). For the Pt NP samples, the rates were 62, 380, and 440 mol$_{H2}$ Mol$_{Pt}$$^{-1}$ min$^{-1}$, for Pt/graphene-WI, Pt/carbon, and Pt/SiO$_2$, respectively, close to the values for Pt catalysts reported in the literature (Supplementary Table 4). The rate of PtO$_2$ was 197 mol$_{H2}$ Mol$_{Pt}$$^{-1}$ min$^{-1}$. Obviously, hydrolytic dehydrogenation of AB on Pt catalysts is a structure sensitive reaction, the size, and electronic properties of Pt NPs might both play important roles[57,58]. In sharp contrast with the above samples, the Pt$_2$ dimers exhibited the highest rate of 2800 mol$_{H2}$ Mol$_{Pt}$$^{-1}$ min$^{-1}$ ever reported in literature, which was ~17 and 45 times higher than the corresponding single-atom Pt$_1$/graphene and Pt/graphene-WI samples, respectively. When the mole ratio of Pt to the AB substrate was increased, Pt$_2$/graphene could preserve the high specific rate to a large extent (Supplementary Fig. 20). Note that all the Pt samples produced a similar product of BO$_2$$^-$ in the spent reaction solution, according to the identical $^{11}$B resonance at 8.9 ppm (Supplementary Fig. 21)[56].

DFT calculations were further carried out to get a deeper insight into the vast activity difference between Pt$_1$/graphene and Pt$_2$/graphene. Since AB is known as an excellent reducing agent[59], and could likely stripe off the terminal dioxygen of Pt$_1$/graphene and Pt$_2$/graphene (the insets of Fig. 3d, e) during the reaction, partially reduced structures without the terminal dioxygen were considered for both Pt$_1$/graphene and Pt$_2$/graphene (the insets of Fig. 5a). The reduced samples are denoted as Pt$_1$/graphene-R and Pt$_2$/graphene-R, respectively. First, we compared the projected density of states of the 5d orbitals of the Pt atom in Pt$_1$/graphene-R and the top Pt atom in Pt$_2$/graphene-R. It was found that the unoccupied 5d states of the top Pt atom in Pt$_2$/graphene-R locates at a considerably higher energy position of 0.87 eV above Fermi level ($E_f$) than that of the Pt atom in Pt$_1$/graphene-R (0.40 eV), which indicates that Pt$_1$/graphene-R is more prone to accept electrons than Pt$_2$/graphene-R (Fig. 5a). This result is in line with the recent literature where Åstrand et al. reported that Pt$_1$ single atom had a more strong ability to accept electrons than the top Pt atom in Pt$_2$ dimer, thereby showing stronger CO adsorption on Pt$_1$[60,61].

When AB is adsorbed on Pt$_1$/graphene-R, two B–H bonds were significantly elongated from 1.22 to 1.42 Å, with a strong adsorption energy of −3.20 eV (Fig. 5b). On Pt$_2$/graphene-R, the elongation of the two B–H bonds was slightly less, to 1.39 Å, and the AB adsorption energy was considerably weaker, about −2.81 eV (Fig. 5c). The adsorption of AB on Pt (111) was also investigated as a comparison (Supplementary Fig. 22). We found

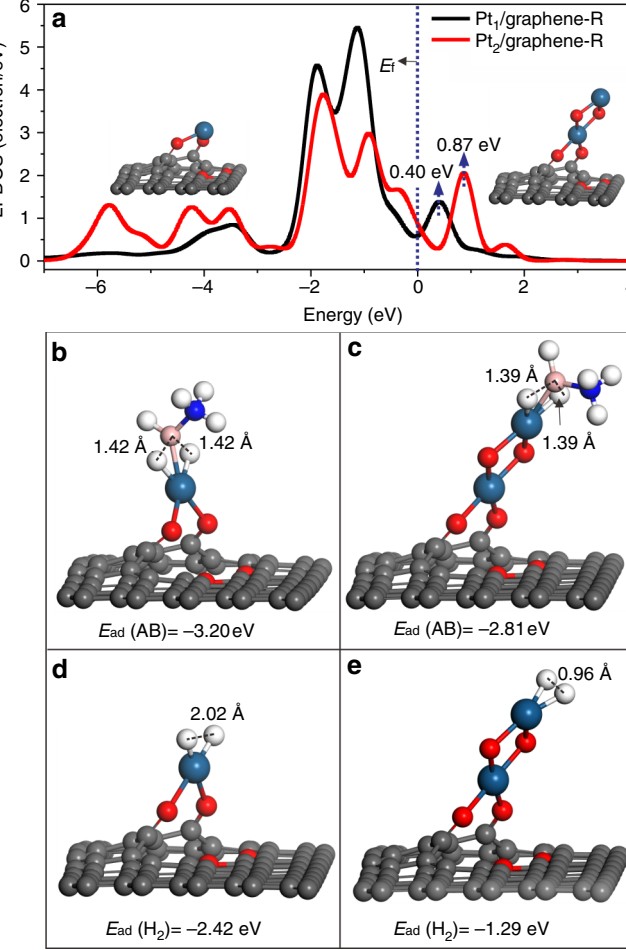

**Fig. 5** Electronic properties as well as AB and H₂ adsorptions. **a** The local partial density of state (LPDOS) projected on the Pt 5d orbitals of Pt₁/graphene-R and the top Pt atom in Pt₂/graphene-R. The local configurations for AB adsorption on Pt₁/graphene-R (**b**), Pt₂/graphene-R (**c**). The local configurations for H₂ adsorption on Pt₁/graphene-R (**d**), Pt₂/graphene-R (**e**). The ball in gray, white, pink, blue, red, and dark blue represent carbon, hydrogen, boron, nitrogen, oxygen, and platinum, respectively

that AB could quickly dissociate to three H atoms and BNH₃ species without any barrier. The AB adsorption energy is −3.97 eV, significantly stronger than that on Pt₁ single atom and Pt₂ dimer. The strong AB adsorption on Pt (111) revealed by DFT calculations agrees well with the literature[57], where Pt NP catalyst deactivation induced by B poisoning was observed during the AB hydrolysis reaction. Bearing this in mind, we further examined the recycling stabilities of the Pt₁/graphene, Pt NP, and PtO₂ catalysts and measured the B contents in the used samples using ICP-AES. Indeed, catalyst deactivations and considerable amounts of B adsorption were observed on all the used samples (Supplementary Figs. 23 and 24), in line with the literature[57]. In addition, sintering and leaching of Pt were also noticed on Pt₁/graphene, Pt/graphene-WI, and Pt/SiO₂ (Supplementary Fig. 25). Therefore, the considerable weaker adsorption of AB on Pt₂ dimer could be one key factor for its high activity as shown in Fig. 4.

H₂ adsorptions on Pt₁/graphene-R and Pt₂/graphene-R were also investigated as a descriptor of hydrogen desorption from the catalyst surface during the AB hydrolysis reaction (Fig. 5d, e). It was again found that H₂ adsorption on Pt₁/graphene-R

(−2.42 eV) is remarkably stronger than that on Pt₂/graphene-R (−1.29 eV). More interestingly, we found that H₂ chemisorbs dissociatively on Pt₁/graphene-R, but molecularly on Pt₂/graphene-R, indicated by the H–H bond distance of 2.02 and 0.96 Å, respectively. Such molecular adsorption of H₂ on the Pt₂ dimer with a moderate adsorption energy favors H₂ desorption during the AB hydrolysis reaction, thereby further boosting the catalytic activity. Taken together, compared to Pt₁ single atom, the higher energy position of the unoccupied state of the Pt 5d orbital of the top Pt in the Pt₂ dimer might play an important role in weakening the adsorption of both AB and H₂ molecules, thus facilitating the activity remarkably.

**Stability of Pt₂ dimers on graphene**. In sharp contrast with all other Pt samples shown in Supplementary Fig. 23, the dimeric Pt₂/graphene catalyst exhibited a very high stability in the AB hydrolysis reaction during the recyclability test for five cycles (Fig. 6a). STEM measurements showed that there was no any visible Pt NPs formation and Pt₂ dimers remained as the main features in the used sample (Fig. 6d, g). The Pt₁ single atoms shown in Fig. 6g were likely produced by the electron beam during STEM measurements (Supplementary Figs. 7–9), since there was no apparent activity decrease. When Pt₂/graphene was annealed at high temperatures in helium, the activity declined considerably (Fig. 6b), but rates still remained as high as 1670 and 1037 mol$_{H_2}$ Mol$_{Pt}^{-1}$ min$^{-1}$ for the sample annealed at 300 (Pt₂/graphene-300C) and 400 °C (Pt₂/graphene-400C), respectively (Fig. 6c). Obviously, significant amounts of Pt₂ dimers were survived after the high-temperature treatments. Indeed, HAADF-STEM revealed a mixture of Pt₂ dimers, Pt₁ single atoms, and Pt NPs in both Pt₂/graphene-300C (Fig. 6e, h) and Pt₂/graphene-400C (Fig. 6f, i). Please keep in mind that these STEM images might significantly underestimate the portion of Pt₂ dimers in the samples owing to the possible beam damage during STEM measurements (Supplementary Figs. 7–9).

In conclusion, we have successfully demonstrated that Pt₂ dimers can be bottom–up constructed on graphene with a high-surface area. We found that the type of surface nucleation sites, selective deposition, the self-limiting nature of ALD, and the high stabilities of Pt₁ single atoms and Pt₂ dimers are the keys factors for the Pt₂ dimers synthesis. The dominant presence of Pt₂ dimers on graphene in the oxidized form of Pt₂O$_x$ were confirmed by a combination of aberration-corrected HAADF-STEM, ICP-AES, and XAFS and DFT calculations. Rotating and uncoupling of Pt₂ dimers under the electron beam during STEM measurements, provide direct evidence of the presence of Pt₂ dimers on graphene. In the AB hydrolysis reaction, the dimeric Pt₂/graphene catalyst exhibited a strikingly high activity, which was ~17- and 45-fold higher than graphene supported Pt₁ single atoms and Pt NPs, respectively. The lower adsorption energies of AB and H₂ on the Pt₂ dimers than that on Pt₁ single atoms or Pt NPs are likely the major reasons for the high activity. Importantly, the dimeric Pt₂/graphene catalyst showed a high stability under the current reaction conditions and in the inert environment at below 300 °C. Finally, our findings point out a new avenue to bottom–up synthesis of atomically precise ultrafine metal (and/or metal oxide) clusters on high-surface area supports for advanced catalysis.

## Methods

**Materials**. Trimethyl(methylcyclopentadienyl)platinum(IV) (MeCpPtMe₃, 98%), chloroplatinic acid (H₂PtCl₆, ≥99.9%, trace metals basis), ammonia borane (97%), the commercial PtO₂ (≥99.9%, 70 m²/g), and Pt/carbon catalysts (the Pt content, 5.0 wt%) were all purchased from Sigma Aldrich. Silica gel was purchased from Alfa Aesar (Brunauer, Emmett, and Teller (BET) surface area 300 m²/g). Pristine graphene nanosheet (99.5%) was bought from Chengdu Organic Chemicals Co.

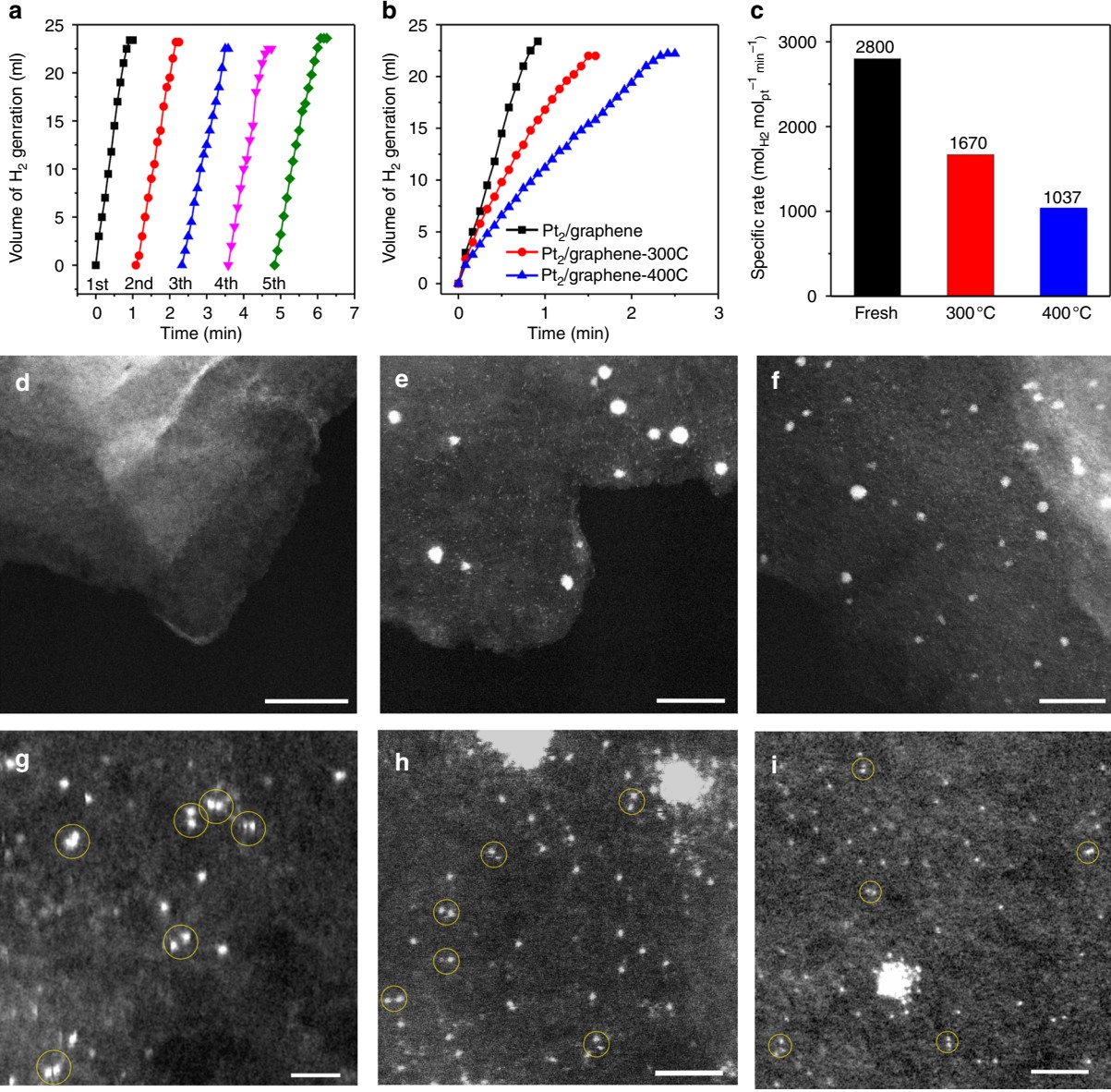

**Fig. 6** Stability of the dimeric $Pt_2$/graphene catalyst. **a** Five recycles in hydrolytic dehydrogenation of AB at room temperature over the dimeric $Pt_2$/graphene catalyst by adding additional 0.325 mmol of pure AB into the reaction flask after each run. **b** Plots of time vs volume of hydrogen gas generated from AB hydrolysis and **c** the corresponding specific rates at room temperature over the dimeric $Pt_2$/graphene catalysts after different pretreatments: as-prepared, annealing in helium at 300 and 400 °C for 1 h, respectively. **d**, **g** Representative HAADF-STEM images of the used $Pt_2$/graphene catalyst after the recyclability test, scale bars, 20 nm (**d**), 1 nm (**g**). **e**, **h** Representative HAADF-STEM images of the $Pt_2$/graphene catalyst after annealing in helium at 300 °C for 1 h, scale bars, 10 nm (**e**), 2 nm (**h**). **f**, **i** Representative HAADF-STEM images of the $Pt_2$/graphene catalyst after annealing in helium at 400 °C for 1 h, scale bars, 10 nm (**f**), 2 nm (**i**). $Pt_2$ dimers in **g–i** are highlighted by yellow circles

Ltd., Chinese Academy of Sciences. All materials were used as received without further purification.

**Preparation of reduced graphene oxide**. Pristine graphene nanosheet was first oxidized to graphene oxide according to the procedure described previously[62]. In brief, 0.6 g graphene nanosheet and 0.3 g sodium nitrate was sequentially added into concentrated sulfuric acid ($H_2SO_4$, 15 mL) and stirred at room temperature for 22 h. Then the mixture was cooled down to 0 °C to add 1.8 g potassium permanganate ($KMnO_4$). After stirring at room temperature and 35 °C for 2 and 3 h, respectively, the mixture was heated to 98 °C and kept at this temperature for another 30 min. Next, it was cooled down to 40 °C, and 90 mL of water and 7.5 mL of hydrogen peroxide ($H_2O_2$, 30%) were slowly added into the mixture. After that the precipitate was filtered out by washing with HCl (5%) and ultrapure water, it was dried in a vacuum oven at 45 °C overnight. The dry material was grinded to obtain graphene oxide powder. Finally, the reduced graphene oxide was obtained by thermal deoxygenation of graphene oxide powder at 1050 °C for 2 min under helium at a flow rate of 50 mL/min.

**Synthesis of $Pt_1$/graphene**. Pt ALD was carried out on a viscous flow reactor (GEMSTAR-6 Benchtop ALD, Arradiance) by alternatively exposing to MeCpPtMe₃ precursor and $O_2$ (99.999%) at 250 °C[46,63,64]. Ultrahigh purity $N_2$ (99.999%) was used as the carrier gas at a flow rate of 200 mL/min. The Pt precursor was heated to 65 °C to get a sufficient vapor pressure. The reactor inlets were held at 110 °C to avoid any precursor condensation. The timing sequence was 90, 120, 60, and 120 sec for the MeCpPtMe₃ exposure, $N_2$ purge, $O_2$ exposure, and $N_2$ purge, respectively (90-120-60-120).

**Synthesis of $Pt_2$/graphene**. The second Pt ALD cycle was performed on the $Pt_1$/graphene SAC at 150 °C. Here, $O_3$ was used as the oxidant to make sure that the precursor ligand can be fully removed[65]. The timing sequence was 90-120-60-120.

**Synthesis of 2cPt/graphene**. Two consecutive cycles of Pt ALD was also performed on the reduced graphene oxide at 250 °C using the same timing sequence.

**Synthesis of Pt/SiO₂**. Pt ALD was performed on the silica gel support for one cycle at 250 °C using the same timing sequence.

**Synthesis of Pt/graphene-WI**. A Pt/graphene NP catalyst was synthesized by the wetness impregnation method (Pt/graphene-WI). In this case, 100 mg graphene support was slowly added into a $1.93 \times 10^{-2}$ M $H_2PtCl_6$ aqueous solution (0.9 mL). Then, the mixture was stirred for 30 min, and then dried in air at room temperature for 12 h. The dried material was first calcined in air at 120 °C for 12 h, and then reduced in 10% $H_2$ in argon at 300 °C for another 2 h to get the Pt/graphene-WI catalyst.

**Catalyst characterization**. The Pt loadings in these samples were determined by ICP-AES measurements; therein all samples were dissolved in hot fresh aqua regia. The BET surface area was measured on a Micromeritics ASAP 2020 system. Raman spectra were recorded on a LabRAM HR Raman spectrometer with a 514 nm Ar laser in backscattering geometry. Aberration-corrected HAADF-STEM measurements were taken on a JEM-ARM200F instrument (University of Science and Technology of China) at 200 keV. XAFS measurements at the Pt $L_3$-edge (11,564 eV) were performed in the transmission mode with the Si (111) monochromator at the BL14W1 beamline of the Shanghai Synchrotron Radiation Facility (SSRF), China. The storage ring of SSRF worked at 3.5 GeV with a maximum current of 210 mA.

**XAFS data analysis and simulation**. The acquired EXAFS data were processed according to the standard procedures using the ATHENA module implemented in the IFEFFIT software packages[61]. The EXAFS oscillation functions $\chi(k)$ were obtained by subtracting the post-edge background from the overall absorption spectra and then normalized with respect to the edge-jump step. The $R_{bkg}$ value of 1.0 was used for all samples. Subsequently, $k^3$-weighted $\chi(k)$ fucntions in the $k$ range of 2.2–13.5 Å⁻¹ were FT to the $R$ space by using a Hanning window of $dk = 3.0$ Å⁻¹.

EXAFS simulations were performed with the FEFF8.4 code[62] using the structural models suggested by DFT calculations. The simulated EXAFS $\chi(k)$ functions were also $k^3$-weighted and FT into the $R$-space by using the same $k$ range of 2.2–13.5 Å⁻¹ as that in the experimental data. During simulations, the coordination numbers were set to the values of the model structures generated by DFT calculations. The amplitude reduction factor $S_0^2$ was fixed at the value of 0.86 which was determined by fitting the reference metal Pt foil. The Debye–Waller factors for the nearest Pt–O/C and Pt–Pt pairs were set at the typical values of 0.0030 and 0.0065 Å² determined from the fittings of $PtO_2$ and Pt foil references, respectively, and they were set at 0.008 Å² for all the other distant paths which contributed barely discernible signals as seen from the experimental data in the $R$-space. To further improve the match between the simulation and the experimental data, for the MeCpPtMe/graphene sample, the two nearest Pt–C and Pt–O interatomic distances were optimized to 2.00 and 2.02 Å, respectively, both of which are within ~ 2% error level as compared to the optimized structure by DFT calculations (2.05 and 2.06 Å, respectively). For the other two samples, the simulated EXAFS spectra based on the DFT-generated structures match well with the experimental data, thus no further structure optimization was performed during EXAFS simulations.

**DFT Calculations**. All spin-polarized calculations were performed by using the DFT method. The DFT Semi-core Pseudopotential method[66] with a double numerical basis set together with polarization functions (DNP) were adopted to form the Perdew-Burke-Ernzerhof (PBE) exchange-correlation functional within the generalized gradient approximation[67], implemented in DMol³ package (DMol³ is a density functional theory quantum mechanical package available from Accelrys Software Inc.)[68]. A DFT-D semi-empirical correction with Tkatchenko-Scheffler (TS) method is applied with the PBE functional to account for the dispersion interaction. Conductor-like screening model (COSMO) with a dielectric constant of 78.54 is adopted to consider the water solvent effect regarding the adsorption of molecule and fragment in AB hydrolysis. A smearing of 0.001 Ha to the orbital occupation is applied to achieve electronic convergence. The real-space global cutoff radius is set to be 4.5 Å. A hexagonal supercell containing (8 × 8) unit cells of graphene monolayer with about 17 Å vacuum layer was used as a support. The convergence tolerances of energy, force, and displacement for the geometry optimization were $1 \times 10^{-5}$ Ha, 0.002 Ha/Å, and 0.005 Å, respectively. $1 \times 1 \times 1$ $k$-points grid is used to describe the Brillouin zone for geometric optimization. The adsorption energy is defined by the formula: $E_{ads(AB)} = E_{(AB/catalyst)} - (E_{catalyst} + E_{AB})$ and $E_{ads(H2)} = E_{(H2/catalyst)} - (E_{catalyst} + E_{H2,gas})$, where $E_{(AB/catalyst)}$, $E_{(H2/catalyst)}$, and $E_{catalyst}$ are the total energies for the optimized equilibrium configurations of catalyst with and without AB or $H_2$, respectively; and $E_{AB}$ ($E_{H2,gas}$) is the energy of the AB (gas phase $H_2$) molecule in its ground state. For Pt(111), the supercell is (4×4).

**Hydrolytic dehydrogenation of AB**. As a probe reaction, hydrolytic dehydrogenation of AB was performed in a three-necked flask at 27 °C under atmospheric pressure. The flask was immersed in a water bath to control the reaction temperature. 10 mg of the $Pt_2$/graphene catalyst was used, while the weight of other catalysts was adjusted to keep the same amount of Pt with $Pt_2$/graphene. Typically, 5 mL aqueous AB solution ($6.5 \times 10^{-2}$ M) were introduced into the glass container via a syringe. For the commercial $PtO_2$ catalyst, the mole ratio of Pt to AB was kept as the same with other samples, and the result was normalized to other samples based the amount of Pt. The AB solution and the catalyst were well-mixed by using a magnetic stirrer at a speed of 800 r/min to eliminate any mass-transfer issue. The generated volume of $H_2$ was measured by a water-filled gas burette, where the volume of water discharged was converted into the volume of hydrogen generated[69].

The specific rates ($r$) of these catalysts were calculated according to the Eq. (2):

$$r = \frac{n_{H2}}{n_{Pt} \times t} \qquad (2)$$

Here $n_{H2}$ is the mole of generated $H_2$, while $n_{Pt}$ is the total mole of Pt in the sample. $t$ is the reaction time in min.

**Data availability**. All the relevant data are available from the authors upon request.

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

## Acknowledgements

This work was supported by the National Natural Science Foundation of China (21473169, 21673215, 21533007, and 21233007), Innovative Research Groups of the National Natural Science Foundation of China (11621063), the One Thousand Young Talents Program under the Recruitment Program of Global Experts, the Young Scientists Fund of the National Natural Science Foundation of China (11404314), Anhui Provincial Natural Science Foundation (1708085MA06), and the Fundamental Research Funds for the Central Universities (WK2060030017). The calculations were performed on the supercomputing system in USTC-SCC and Guangzhou-SCC. We gratefully thank the BL14W1 beamline at the Shanghai Synchrotron Radiation Facility (SSRF), China.

## Author contributions

J.L. conceived the idea and designed the experiments. H.Y. performed catalyst synthesis and catalytic performance evaluations. Y.L. performed the STEM measurements. S.W., Z.S., H.C., W.L., and T.Y. performed the XAFS measurements. W.Z., H.W., and J.Y. performed the DFT calculations. C.W., J.L., and X.H. assisted catalyst characterization and catalytic performance tests. J.L. wrote the manuscript. All the authors contributed and commented on the manuscript. H.Y. and Y.L. contributed equally to this work.

## Additional information

**Competing interests:** The authors declare no competing financial interests.

