## [Peer Review File · Nature Communications]

Reviewer #1 (Remarks to the Author):

The paper deals with the synthesis of Pt dimers by atomic layer deposition. The catalysts have been carefully characterized by various techniques such as Aberration-corrected scanning transmission electron microscopy, x-ray absorption spectroscopy. The graphene supported Pt dimers exhibited a high turnover frequency of 2800 molH₂ molPt⁻¹ min⁻¹ at room temperature. The results are very interesting in terms of synthesis and applications of precise controlled Pt clusters. However, several issues need to be addressed before publication.

- 1) The work focused the new catalyst of Pt dimers on graphene. The results also clearly show a support dependent activity of single atom Pt catalysts, though it is not discussed. The SiO₂ supported single Pt has a much higher activity than graphene supported single Pt. The properties of the support seems to be important here. However, the properties of the graphene were not reported here. In fact, it is not convinced here the support being single layer or multilayer graphene. The graphene is very easily restacked to graphite. High resolution TEM showed flacks of graphite, or multilayer graphene. However, the electronic properties of graphene depend significantly on the layer numbers. The surface area of the support need to be reported.
- 2) The cluster is identified as Pt₂O₆. It is not reported that the catalyst active sites are Pt or Pt₂O₆.
- 3) The Pt loading of all they catalysts should be reported.
- 4) It has been shown that dehydrogenation of ammonia borane is significantly structure sensitive (J Am Chem Soc, 136 (2014) 16736-16739). The results here are clearly different from the literature results. A discussion on the specifically high activity of Pt dimmer is needed.

Reviewer #2 (Remarks to the Author):

This paper reports a “bottom-up” fabrication of Pt dimers on graphene through a ALD method. A claim was made for the selective deposition of a Pt atom on a preliminary one using ALD. The dimeric Pt₂/graphene catalyst showed a high stability in the environment at below 300 °C. In hydrolytic dehydrogenation for hydrogen generation, the dimeric Pt₂/graphene catalyst exhibited a strikingly high activity, which was much higher than that of graphene supported Pt single atoms and nanoparticles. Unfortunately, the enhanced mechanism is not mentioned in the manuscript. I think

this paper is not publishable in its current state, unless the key effect for the extremely enhanced property is clearly demonstrated. Some questions and suggestions are listed below.

1. If the dimers are successfully prepared, the authors can try to prepare Pt trimers and tetramers based on the growth mechanism. The comparison of the catalytic properties of dimers, trimers and tetramers might show the key role for the enhanced dehydrogenation activity.
2. For hydrogen production, Pt is also widely used as effective catalyst in electrochemical hydrogen evolution reaction. Can the Pt dimer catalysts exhibit the similar properties in electrochemical catalytic reactions?
3. Why Pt single atoms can form during the STEM measurements?
4. As mentioned in the ms, the formed Pt dimers is Pt₂O_x instead of pure Pt metal. It should be more reasonable to compare the catalysts with commercial PtO₂.
5. The authors claim that mono-nucleation of MeCpPtMe₃ at one nucleation site during precursor exposure occurs exclusively at phenol or phenol-carbonyl pairs. However, a paper published by Kim et al in demonstrated the selective deposition of Pt by ALD at graphene line defects. Is it not possible that under the deposition conditions employed that Pt may also be depositing at line defects of edge carbon?
6. The authors claim that the second exposure of Pt, at a lower temperature, selectively deposits on the Pt atom deposited during the first cycle. This claim is backed up by ICP-AES data as well as SEM images. However, the authors have not experimented with Pt pulse time. If they authors do claim that saturation has been achieved, then extending the pulse time of Pt, should result in similar ICP-AES values.
7. Line 124 describes a Pt ALD experiment on reduced graphene oxide demonstrating the formation of Pt single atoms. If the authors claim selective deposition to phenol or phenol-carbonyl pairs, what is the Pt anchoring onto in this experiment? Furthermore, are these anchoring sites still present in the selectively reduced graphene oxide substrate?
8. An observation is stated in the manuscript regarding the rotation of Pt dimers to either angles of 30, 60 or 90 under the electron beam. The author's claim that this observation is unambiguous evidence for the presence of Pt dimers. What is the evidence for this? How does this observation disregard the possibility of projection coincidence?
9. Figure 3c,d,e displays EXAFS spectra for various experiment. The initial peak rise for the experimental data appears to be suspiciously pinched at R values below 1. This is often an indication of a high R_{bkg} being used. What was the R_{bkg} employed and what is the justification of using this value?
10. Figure 3b shows EXAFS data for standard samples as well as samples prepared by ALD as well as the Pt precursor itself. The regularly spaced ripples for the MeCpPtMe₃ suggests that an appropriate Dr or Dk was not applied. What k window was used and what FT operation function was

employed. These factors will affect how the FT is applied to the discrete sign wave and will significantly affect the EXAFS data.

11. The reviewer would like to know the justification for using a k3 weight for FT EXAFS data? Typically when EXAFS data is analyzed for lighter elements, a k2 weight is employed. What is the justification for using a higher k weight?

12. Supplementary figure 10 displays k3 weight EXAFS oscillations and corresponding EXAFS simulations for different ALD experiments. Many of the simulation curves presented do not match up at all with the experimental. What is the reason for this? Furthermore, were Debye-Waller factors, Coordination number, or R-factors obtained from simulations? What are the best fit values from these simulations compared to your experimental data?

13. There are number of grammatical and spelling errors littered throughout the paper. Ensure someone with an English background carefully looks through the paper before submitting again.

Reviewer #3 (Remarks to the Author):

This paper describes the synthesis and testing of a series of catalysts made via the ALD of Pt precursors on defective graphite. The main achievement is the synthesis of catalysts that consist mainly of Pt dimers, which are argued to be very efficient in the dehydrogenation of ammonia borane. The synthesis of catalysts with atomic dispersions between the single atom and nanoparticle regime remains challenging so this is where the novelty of this paper lies. However, I have some fairly serious concerns about the experiments and data interpretation that should be addressed before publication. Furthermore, this paper needs a thorough edit by a native English speaker. The grammar and flow make reading the manuscript and SI very difficult at present. To name just a few examples of these problems – even the abstract is riddled with errors:

“Therein, one atom variation might...” This doesn’t make sense

“Progresses are made in the...” grammar

“Remains a grant challenge” I assume you mean grand?

“Pt single atoms deposition” should be “atom”

“45- folds...” should read “45-fold”

Note – these are just errors in the abstract – the whole paper and SI should be carefully edited and fixed.

Major Scientific Points:

The biggest issue with the paper is that it is a report of the synthesis of Pt₂ catalysts and results of testing for dehydrogenation activity. The paper lacks clear explanations for:

- 1) the nature of the binding site on graphite – its speculative at present
- 2) the exact nature of the Pt₂ complexes i.e. is C involved
- 3) what about Pt₂ vs. Pt₁ makes these Pt₂ clusters active?
- 4) what is making some of the catalysts deactivate?

The authors have performed DFT modeling of the Pt₂ clusters but don't present any electronic information about this cluster and how it might facilitate catalysis. This s, p, d state density information can be easily pulled from those calculations and should be compared to Pt₁ and Pt nanoparticles so to establish what is unique about the Pt₂ clusters. Furthermore, DFT should be used to calculate molecular binding strengths of the molecule of interest to these structures i.e. Pt₁ vs. Pt₂ vs Pt NPs and the reaction barriers for breaking B-H, N-H bonds and the barrier to H₂ formation quantified. The authors make some hand waving arguments based on the Sabatier principle to explain their findings but these additional DFT calculations would go much further to explaining why their Pt₂ systems works so well.

The main point of the paper is that formation of Pt₂ clusters is self limiting and on page 7 the authors claim that the reason for this is steric. This should be expanded upon as it is only one of several possible explanations that needs stronger support if its to be used as the sole explanation of Pt₂ formation mechanism which is central to the paper.

XAFS – the spectra of the Pt₁ vs Pt₂ in Fig 3 look identical to me and the simulation doesn't match the experimental data well at all. How can such strong conclusions be made from these XAFS spectra?

Page 11 line 305 – I don't understand why changing the mole ratio of the reactant to the catalyst doesn't affect the rate? If you are studying the activity of catalysts you need to use a regime in which

the intrinsic reaction rate dominates in order to make a comparison between catalysts. i.e. you double the catalyst amount you double the rate.

Minor Scientific Point:

Page 10 line 255-6 comparing Pt-Pt to Pt-O distances is not informative, the Pt-Pt distances for the Pt₂ complex should be given instead.

Reviewer #1 (Remarks to the Author):

The paper deals with the synthesis of Pt dimers by atomic layer deposition. The catalysts have been carefully characterized by various techniques such as Aberration-corrected scanning transmission electron microscopy, x-ray absorption spectroscopy. The graphene supported Pt dimers exhibited a high turnover frequency of 2800 molH₂ molPt⁻¹ min⁻¹ at room temperature. The results are very interesting in terms of synthesis and applications of precise controlled Pt clusters. However, several issues need to be addressed before publication.

Response: We appreciate the reviewer's positive comments on our work.

1) The work focused the new catalyst of Pt dimers on graphene. The results also clearly show a support dependent activity of single atom Pt catalysts, though it is not discussed. The SiO₂ supported single Pt has a much higher activity than graphene supported single Pt. The properties of the support seem to be important here. However, the properties of the graphene were not reported here. In fact, it is not convinced here the support being single layer or multilayer graphene. The graphene is very easily restacked to graphite. High resolution TEM showed flakes of graphite, or multilayer graphene. However, the electronic properties of graphene depend significantly on the layer numbers. The surface area of the support needs to be reported.

Response: We appreciate the reviewer's valuable comments.

Firstly, we would like to point out that the 1cPt/SiO₂ catalyst is a Pt nanoparticle (NP) catalyst, but not a single-atom catalyst. We are sorry for the confusion, and we changed the name of 1cPt/SiO₂ to Pt/SiO₂ in the revised manuscript to make it more clear.

In the present work, different reactivities were observed on Pt/SiO₂, Pt/carbon, and Pt/graphene-WI NP catalysts, which might be mainly caused by the following two factors:

- (a) **Different Pt particle sizes.** Hydrolytic dehydrogenation of ammonia borane (AB) is highly structure sensitive. For Pt NP catalysts, the catalytic activity strongly depends on the Pt particle size, according to the literature [Chandra, M., Xu, Q., J. Power Sources 168, 135-142 (2007); Chen, W., et al., J. Am. Chem. Soc. 136, 16736-16739 (2014)]. In our case, the Pt particle sizes were 1.8 ± 0.5, 1.9 ± 0.3, and 2.3 ± 0.7 nm, for Pt/graphene-WI, Pt/SiO₂, and Pt/carbon, respectively. The activities are in the order of Pt/SiO₂ > Pt/carbon > Pt/graphene. This trend is close to the one reported by Chen et al, where the 1.8 nm sized Pt/CNT catalyst showed the highest activity [Chen, W., et al., J. Am. Chem. Soc. 136, 16736-16739 (2014)].
- (b) **Different electronic properties of Pt NPs induced by the support.** The electronic properties of Pt NPs might also have a considerable impact on the catalytic activity. For instance, Chen et al. demonstrated electron-deficient Pt NPs supported on carbon nanotubes had higher activity in hydrolytic dehydrogenation of AB [Chen, W. et al., Chem. Commun. 50, 2142-2144 (2014)].

Regarding the properties of the reduced graphene oxide support, additional characterization using HAADF-STEM and Raman spectroscopy was carried out. We found that the thickness of the reduced graphene oxide was about 9 graphene layers according to the high resolution STEM image in **Fig. R1a**, where the distance between the fringes is about 0.356 nm, consistent well with the distance between the graphene layers, according to the literature [Geim A. K., Grigorieva I. V., *Nature* 499, 420-425 (2013)]. Raman spectroscopy was also carried out on the pristine and reduced graphene oxide support. As shown in **Fig. R1b**, the D-band ($\sim 1345\text{ cm}^{-1}$) to G-band ($\sim 1580\text{ cm}^{-1}$) ratio for the reduced graphene oxide was much higher than that for the pristine graphene. Such high D-band to G-band ratio implies that there were small domains of the sp^2 carbons, along with a large amount of defects on the reduced graphene oxide [Yang, D. X., et al., *Carbon* 47, 145-152 (2009)]. **In brief, the reduced graphene oxide support is defect-rich multilayered graphene films with a thickness of a few nanometers.**

The surface area of the reduced graphene oxide support was about $570\text{ m}^2/\text{g}$, according to the Brunner–Emmet–Teller (BET) surface area measurement.

The above discussion is added into the revised manuscript on pages 4, 11 and 12, Fig. R1 is also added into the Supplementary information as Supplementary Figure 1. The changes are highlighted in yellow.

Fig. R1. (a) A HAADF-STEM image of the reduced graphene oxide support at a high magnification, the inset shows the enlargement of the white rectangular area. (b) The Raman spectra of pristine graphene and reduced graphene oxide.

2) The cluster is identified as Pt_2O_6 . It is not reported that the catalyst active sites are Pt or Pt_2O_6 .

Response: This is an excellent question.

AB is known as an excellent reducing agent [Andrews, G. C. & Crawford, T. C., *Tetrahedron Lett.* 21, 693-696 (1980)], and could likely stripe off the terminal dioxygen in both $\text{Pt}_1/\text{graphene}$ and $\text{Pt}_2/\text{graphene}$ during the reaction (the insets of **Fig. 3d, e**), thus partially reduced structures without the terminal dioxygen were considered for both $\text{Pt}_1/\text{graphene}$ and $\text{Pt}_2/\text{graphene}$. The active site for $\text{Pt}_2/\text{graphene}$ under the AB hydrolysis reaction conditions is suggested to be Pt_2O_4 as shown in **Fig. R2**.

To get a deeper insight into the vast activity difference between Pt₁/graphene and Pt₂/graphene, DFT calculations were further carried out. In brief, we found that the lower adsorption energies of both AB and H₂ molecules on Pt₂ dimers induced by the higher energy position of the unoccupied states of Pt 5*d* orbital above Fermi level, are likely the major reasons for the significantly higher activity. **Please see more details in our response to the comment #5 by the Reviewer #3. The new DFT calculation results are also added in our revised manuscript on pages 11-13, highlighted in yellow.**

Fig. R2. The structure of the Pt₂/graphene catalyst under the AB hydrolysis reaction conditions.

3) The Pt loading of all the catalysts should be reported.

Response: The reviewer might have missed the data. The Pt loadings of all these catalysts had been shown in the **Supplementary Table 3**.

4) It has been shown that dehydrogenation of ammonia borane is significantly structure sensitive (J. Am. Chem. Soc., 136 (2014) 16736-16739). The results here are clearly different from the literature results. A discussion on the specifically high activity of Pt dimer is needed.

Response: We appreciate the reviewer's valuable comments as well as the suggestion about the excellent reference.

In this reference [Chen, W. et al., *J. Am. Chem. Soc.* 136, 16736–16739 (2014)], Chen et al. showed that the 1.8 nm sized Pt/CNT catalyst had the highest activity, and the activity quickly decreased when the Pt particle size was reduced to 1.5 and 1.4 nm (**Fig. R3a**). Following this work, we plotted the activities of the three Pt NP catalysts as a function of the Pt particle size, we observed a similar volcano shape of specific rates as a function of Pt particle size (**Fig. R3b**), where the maximum activity was observed on Pt/SiO₂ with a Pt particle size of 1.9 ± 0.3 nm.

However, in the subnanometer range, the catalytic properties of ultrafine metal clusters can change largely from metal NPs. When a metal cluster consists of only a few metal atoms, it could have a discrete energy band structure, tightly correlated with the number of metal atoms. Changing one atom in the cluster might largely alter the electronic structure, thus drastically changing their catalytic properties. Such "atom-dependent" catalytic behaviors have been unambiguously demonstrated on the model catalysts of mass-selected metal clusters [Kaden, W. E., Wu, T. P., Kunkel, W. A. & Anderson, S. L., *Science* 326, 826-829 (2009); Palmer, R. E., Pratontep, S. & Boyen, H. G., *Nat Mater* 2, 443-448 (2003); Li, Z. Y. et al., *Nature* 451, 46-U2 (2008); Abbet, S. et al., *J. Am. Chem. Soc.* 122, 3453-3457 (2000); Yoon, B. et al., *Science* 307, 403-407 (2005); Nesselberger, M. et al., *Nat. Mater.* 12, 919-924 (2013); Vajda, S. et al., *Nat. Mater.* 8, 213-216 (2009)].

Therefore, it is not surprising that Pt₁ single atoms and Pt₂ dimers showed sharp differences from each other and from Pt NPs in the AB hydrolysis reaction. For Pt₂ dimers, our new DFT calculations revealed that the lower adsorption energies of both AB and H₂ molecules on Pt₂ dimers are likely the major reasons for the significantly higher activity. **Please see more details in our response to the comment #5 by the Reviewer #3. The new DFT calculation results are also added in our revised manuscript on pages 11-13, highlighted in yellow.**

Fig. R3. (a) Initial hydrogen generation rate ($r_{initial}$) as a function of Pt particle size. Reprinted from [Chen, W., et al., *J. Am. Chem. Soc.* 136, 16736–16739 (2014)]. (b) Initial specific rates of the Pt/SiO₂, Pt/Carbon, and Pt/graphene-WI nanoparticle catalysts in the present work.

Reviewer #2 (Remarks to the Author):

This paper reports a “bottom-up” fabrication of Pt dimers on graphene through a ALD method. A claim was made for the selective deposition of a Pt atom on a preliminary one using ALD. The dimeric Pt₂/graphene catalyst showed a high stability in the environment at below 300 °C. In hydrolytic dehydrogenation for hydrogen generation, the dimeric Pt₂/graphene catalyst exhibited a strikingly high activity, which was much higher than that of graphene supported Pt single atoms and nanoparticles. Unfortunately, the enhanced mechanism is not mentioned in the manuscript. I think this paper is not publishable in its current state, unless the key effect for the extremely enhanced property is clearly demonstrated. Some questions and suggestions are listed below.

Response: We appreciate the reviewer’s positive comments on our work.

1) If the dimers are successfully prepared, the authors can try to prepare Pt trimers and tetramers based on the growth mechanism. The comparison of the catalytic properties of dimers, trimers and tetramers might show the key role for the enhanced dehydrogenation activity.

Response: We appreciate the reviewer’s constructive suggestion. We had tried to prepare Pt trimers. However, there was an issue about selective deposition during the third ALD cycle for synthesis of Pt₃ trimers.

During the synthesis of the dimeric Pt₂/graphene catalyst, ozone was used to remove the Pt precursor ligands at 150 °C in the 2nd cycle of Pt ALD. However, we noticed that the ozone can create additional nucleation sites along with the Pt ligands removal. As a consequence, performing another cycle of Pt ALD on Pt₂/graphene (Pt₃/graphene) would cause an additional Pt nucleation on the new nucleation sites and form a mixture of Pt trimers, dimers and single atoms. Indeed, HAADF STEM showed that there were mixtures of Pt monomers, dimers and trimers on a Pt₃/graphene sample (Fig. R4b). However, it is worthy noting that there was no any visible Pt clusters/nanoparticles in this sample (Fig. R4a,b).

Fig. R4. HAADF-STEM images of the Pt₃/graphene sample at low (a) and high (b) magnifications. (c) The oxygen contents in the reduced graphene oxide (Graphene), the Graphene exposed to O₂ at 250 °C for 10 min (Graphene-O₂) and the Graphene exposed to O₃ (Graphene-O₃) at 150 °C for 10 min, determined by XPS. (d) Plots of time vs volume of hydrogen gas generated from AB hydrolysis at room temperature over the Pt₁/graphene, Pt₂/graphene and Pt₃/graphene catalysts. Notes: some of the Pt single atoms, dimers and trimers in (b) are highlighted by white, yellow and red cycles, respectively.

Here, creation of additional nucleation sites on graphene by ozone was confirmed by XPS (Fig. R4c). XPS measurements showed that the atomic percentage of oxygen in the reduced graphene oxide support was 10.7%, it maintained almost unchanged after O₂ pretreatment at 250 °C for 10 min (graphene-O₂), but increased considerably to 15.4% after O₃ pretreatment at 150 °C for 10 min (graphene-O₃).

The activity of the Pt₃/graphene sample was also evaluated in the AB hydrolysis reaction. Pt₃/graphene showed a reaction rate of 1434 mol_{H₂} Mol_{Pt}⁻¹ min⁻¹, between that for Pt₁ single atoms and Pt₂ dimers (Fig. R4d).

Such non-selective deposition in the third ALD cycle makes the resulting materials nonuniform. Considering the nonuniformity and lower catalytic activity of the Pt₃/graphene catalyst, we will mainly focus on the Pt₂ dimer catalyst in this work. The above discussion is added into the revised manuscript on page 7, highlighted in yellow.

2) For hydrogen production, Pt is also widely used as effective catalyst in electrochemical hydrogen evolution reaction. Can the Pt dimer catalysts exhibit the similar properties in electrochemical catalytic reactions?

Response: The main focus of this work is “bottom-up” synthesis of Pt dimers on a high-surface-area support and its catalytic performance in the AB hydrolysis reaction. Although it could be very interesting to investigate their electrochemical performance, we believe it is out of the focus of this manuscript. While we will certainly investigate it in the near future.

3) Why Pt single atoms can form during the STEM measurements?

Response: This is an excellent question.

It is well-known that electron beam can certainly induce the morphology changes of metal clusters, NPs or even nanorods. Such electron induced morphology changes are often so-called beam damages, and have been observed frequently in many systems [Smith, D. J., Petford-Long, A. K., Wallenberg, L. R., Bovin, J.-O., *Science* **233**, 872-874 (1986). Wang, Z. W., Palmer, R. E., *Nanoscale* **4**, 4947-4949 (2012). Lacroix, L. M.; Arenal, R., Viau, G. *J. Am. Chem. Soc.* **136**, 13075-13077 (2014)].

For example, Lacroix et al. showed that with increase of electron beam exposure time, the electron beam induced the Au nanorod breakdown; and a linear gold atom chain was formed (Fig. R5b), along with a formation of a number of Au single atoms around the rod (Fig. R5c) [Lacroix, L. M., Arenal, R., Viau, G. *J. Am. Chem. Soc.* **136**, 13075-13077 (2014)]. Please also see the impressive videos provided in the supporting information of this reference online.

Therefore, it is not surprising to observe the split of Pt₂ dimers into single atoms by the strong electron beam.

Fig. R5. Electron-beam induced Au nanowire breaking which proceeds through the decrease of the channel diameter from (a) three atoms to (b) a single atom. (c) The resulting 7 nm rods surrounded by Au atoms and dimers. Insets: schematic 3D views of (a) multi-atom-thick chains,

(b) single-atom thick chains, and (c) dimer and isolated atoms. Scale bar = 1 nm. Reprinted from [Lacroix, L. M.; Arenal, R.; Viau, G. D., *J. Am. Chem. Soc.* 136, 13075-13077 (2014)].

4) As mentioned in the ms, the formed Pt dimers is Pt_2O_x instead of pure Pt metal. It should be more reasonable to compare the catalysts with commercial PtO_2 .

Response: We appreciate the reviewer's constructive suggestion.

The catalytic performance of the commercial PtO_2 (Sigma Aldrich, 75 m^2/g) powder was further evaluated in the AB dehydrogenation reaction. We found that PtO_2 generated about 21 mL H_2 in 15 min, with a specific rate of $197 \text{ mol}_{\text{H}_2} \text{ mol}_{\text{Pt}}^{-1} \text{ min}^{-1}$ (Fig. R6). The rate agrees well with the literature [Chandra, M., Xu, Q., *J. Power Sources* 156, 190–194 (2006)].

The new data of PtO_2 is combined into Figure 4 in the revised manuscript. The above discussion is also added accordingly on page 11, highlighted in yellow.

Fig. R6. A plot of time vs normalized volume of hydrogen gas generated from the AB hydrolysis reaction over the commercial PtO_2 powder. The inset shows the specific rate over PtO_2 based on the mole of Pt.

5) The authors claim that mono-nucleation of MeCpPtMe_3 at one nucleation site during precursor exposure occurs exclusively at phenol or phenol-carbonyl pairs. However, a paper published by Kim et al in demonstrated the selective deposition of Pt by ALD at graphene line defects. Is it not possible that under the deposition conditions employed that Pt may also be depositing at line defects of edge carbon?

Response: In Kim's work, Pt ALD was performed at 300 °C [Kim, K., et al., *Nat. Commun.* 5, 4781-4789 (2014)]. However, in our work, Pt ALD was carried out at 250 °C for the first cycle, and 150 °C for the second cycle.

In order to get insight into the temperature effect on the Pt nucleation at the graphene defect sites, the graphene support was first reduced at 1050 °C for 5 min to remove the phenol or phenol-carbonyl pairs groups from the graphene surface (Please see more details in our response to the following comment 7#). Based on the high D-band to G-band ratio in the Raman spectrum of the reduced graphene oxide shown in Fig. R1b, this graphene support contains small domains of the sp^2 carbons, along with a large amount of defects.

According to our previous results shown in **Supplementary Table 1**, one cycle of Pt ALD on this deeply-reduced graphene support would result in a negligible Pt nucleation, although a large amount of defects are present in this graphene revealed by Raman spectroscopy (**Fig. R1b**). Indeed, we found that the Pt loading determined by ICP-AES, was again only 0.02 wt% after performing one cycle of Pt ALD at 250 °C (**Table R1**). Therefore, Pt does not nucleate at the graphene defect sites at 250 °C. However, when Pt ALD was performed on the same graphene support for one cycle at 300 °C, we found that the Pt loading was as high as 2.4 wt% (**Table R1**).

Clearly, the nucleation of Pt at the graphene defect sites was feasible at 300 °C, in a good agreement with Kim's work, but was significantly suppressed at 250 °C or below. This is understandable, the similar temperature effect was also observed by Elam et al., where metal ALD on oxides at lower temperatures was inhibited [Lu, J., Low, K.-B., Lei, Y., Libera, J. A., Nicholls, A., Stair, P. C. Stair, and Elam, J.W., *Nat. Commun.* 5, 3264, (2014)].

The above discussion is added into the revised manuscript on page 5, highlighted in yellow. The new data is included into Supplementary Table 1, and the two references are also added.

Table R1. A comparison of Pt loadings of 1cPt/graphene samples synthesized on the same deeply-reduced graphene support at both 250 and 300 °C, which were determined by ICP-AES.

Sample	Pt ALD temperature (°C)	Pt loading (wt%)
1cPt/graphene	250	0.02
1cPt/graphene	300	2.4

6) The authors claim that the second exposure of Pt, at a lower temperature, selectively deposits on the Pt atom deposited during the first cycle. This claim is backed up by ICP-AES data as well as STEM images. However, the authors have not experimented with Pt pulse time. If they authors do claim that saturation has been achieved, then extending the pulse time of Pt, should result in similar ICP-AES values.

Response: We had examined the MeCpPtMe₃ pulse time. We found that the Pt loading on the graphene became constant, once the MeCpPtMe₃ pulse time was greater than ~ 60 sec (**Fig. R7**). Therefore, the surface reaction during Pt pulse would be saturated with a Pt pulse time above ~60 sec. In our work, a pulse time of 90 sec was employed to make sure it was saturated during each Pt ALD cycle.

On the other hand, as we mentioned in the manuscript, the ratio of Pt loading in Pt-MeCpPtMe/graphene to that in MeCpPtMe/graphene (shown in **Fig. 2h**), was very close to 1. This value also strongly indicates that the Pt pulse time in first Pt ALD cycle was long enough for saturation, otherwise there will be additional Pt deposition during the second Pt ALD cycle, giving a Pt loading ratio higher than 1.

We emphasized the saturated deposition in the revised manuscript on page 5 that "... during the saturated MeCpPtMe₃ exposure (Supplementary Figure 2)". The changes are highlighted in yellow. In addition, Fig. R7 is added into the Supplementary information as Supplementary Figure 2.

Fig. R7. The Pt loadings in the 1cPt/graphene samples synthesized using different MeCpPtMe₃ pulse time during Pt ALD, which were determined by ICP-AES.

7) Line 124 describes a Pt ALD experiment on reduced graphene oxide demonstrating the formation of Pt single atoms. If the authors claim selective deposition to phenol or phenol-carbonyl pairs, what is the Pt anchoring onto in this experiment? Furthermore, are these anchoring sites still present in the selectively reduced graphene oxide substrate?

Response: The graphene support was obtained by thermal reduction of graphene oxide powder at high temperatures. In our previous work [Yan, H., et al., *J. Am. Chem. Soc.* 137, 10484-10487 (2015)], we had systematically investigated the evolution of oxygen functional groups on graphene as a function of reduction temperatures and time using X-ray photoelectron spectroscopy (XPS). As shown in **Fig. R8**, the total oxygen contents in the reduced graphene oxides considerably decreased from 12% to 10%, 7.3%, 3.4%, and 1.9%, for the samples reduced at 700 °C for 30 sec, 1050 °C for 1, 2, 5 and 10 min, respectively (**Fig. R8b-f**). In these reduced graphene oxides, the phenolic oxygen was the dominant one, in line with the literature [Ganguly, A., Sharma, S., Papakonstantinou, P., Hamilton, J., *J. Phys. Chem. C* 115, 17009-17019 (2011)].

In the present work, the graphene oxide was reduced at 1050 °C for 2 min using the identical procedure. Therefore, the total oxygen content was about 7.3% and the phenolic oxygen was about 5.7% in the graphene support. One cycle of Pt ALD on this support resulted in a Pt loading of 0.35%, much smaller than the amount of phenolic oxygen (5.7%). This is probably because that most of oxygen was in the bulk of the reduced graphene oxide. Evidently, we found that the reduced graphene oxide support was defect-rich multilayered graphene films with a thickness of a few nanometer (**Fig. R1**).

Once the graphene oxide was reduced at 1050 °C for longer than 5 min, we found Pt did not grow on this defect-rich multilayered graphene film at 250 °C (**Table R1** and **Supplementary Table 1**). Taken together, ICP-AES and XPS measurements strongly confirmed that the Pt was anchored onto the phenol or phenol-carbonyl pairs as illustrated in **Fig. 1** in our manuscript.

Fig. R8. O1s X-ray Photoelectron Spectroscopy (XPS) of the pristine graphene nanosheet (a) and the graphene supports obtained by thermal reduction of graphene oxide at different temperatures and time in argon. (b) Reduction at 700 °C for 30 sec; (c) 1050 °C for 1 min; (d) 1050 °C for 2 min; (e) 1050 °C for 5 min; (f) 1050 °C for 10 min. Notes: The deconvoluted peaks at around 531.08, 532.03, and 533.43 eV are assigned to C=O (oxygen doubly bound to aromatic carbon), C–O (oxygen singly bonded to aliphatic carbon), O_{ph} (phenolic oxygen), respectively. Atomic percentages of the total oxygen and phenolic oxygen are indicated. Image reprinted from [Yan, H., et al., *J. Am. Chem. Soc.* 137, 10484-10487 (2015)].

8) An observation is stated in the manuscript regarding the rotation of Pt dimers to either angles of 30, 60 or 90 under the electron beam. The author's claim that this observation is unambiguous evidence for the presence of Pt dimers. What is the evidence for this? How does this observation disregard the possibility of projection coincidence?

Response: We appreciate the reviewer's constructive comment.

Under the electron beam exposure during STEM measurements, one of the original Pt-O bond at the interface might be broken. Next, the resulting Pt_2O_x species with one Pt-O-graphene interfacial bond might rotate and form a new Pt-O bond on the other side along the carbon defect edge, as illustrated from **A** to **B** in **Fig. R9a**. In this case, the rotation angle of the Pt_2O_x dimer is 30° . Such rotation is more like "walking" along the defect edge, with the interfacial Pt-O bonds as two "legs". Further "walking" for another step will rotate the Pt_2O_x dimer by 90° ("**A**→**C**"). **Note that such breaking the interfacial bond, rotating, and forming a new bond under electron beam are certainly possible, according to our response to the comment 3# by the reviewer 2#.**

The size of carbon vacancy defects might also influence the rotation angle. For instance, "walking" on a carbon vacancy defect with a smaller size will rotate the Pt_2O_x dimer by 30 and 60° , for "one-step walk" and "two-steps walk", respectively (**Fig. R9b**). **Clearly, the rotation angles are related with both the geometry of the graphene support and the size of carbon defect.**

Fig. R9. The speculated rotation of the dimeric Pt_2O_x species along the graphene defect edge under electron beam. (a) "Walking" along the edge of a carbon defect with a larger size. (b) "Walking" along the edge of a carbon defect with a smaller size.

On the other hand, if Pt_2 dimers are just the coincidence of the projection of two Pt_1 single atoms at different "Z" positions, the characteristic rotation to the specific angles would not be expected.

Therefore, the rotations by the specific angles under the electron beam provide strong evidence of the presence of Pt dimers on graphene.

In the revised manuscript on page 6, we stated that “Such characteristic rotation by the specific angles under the electron beam might be related with the geometry of the graphene support and the size of carbon defect by considering the aforementioned structure of Pt₂ dimer (Supplementary Figure 10). This observation provides strong evidence of the presence of Pt₂ dimers instead of the “projection coincidence” from two isolated Pt₁ atoms at different “Z” positions.” Figure R9 is also added into the Supplementary information as Supplementary Figure 10. The changes are highlighted in yellow.

9) Figure 3c,d,e displays EXAFS spectra for various experiment. The initial peak rise for the experimental data appears to be suspiciously pinched at R values below 1. This is often an indication of a high Rbkg being used. What was the Rbkg employed and what is the justification of using this value?

Response: For all our samples, we used the same Rbkg = 1.0 Å, which is a recommended value for the ATHENA module of the IFEFFIT code. This choice is justified by the criterion that the value is less than half the near-neighbor distance of the Pt-C/O pairs (2.02 Å) and it can effectively minimize the signals below Rbkg in the R-space.

10) Figure 3b shows EXAFS data for standard samples as well as samples prepared by ALD as well as the Pt precursor itself. The regularly spaced ripples for the MeCpPtMe₃ suggests that an appropriate Dr or Dk was not applied. What k window was used and what FT operation function was employed. These factors will affect how the FT is applied to the discrete sign wave and will significantly affect the EXAFS data.

Response: We appreciate the reviewer’s valuable comment.

We are sorry for having not giving enough details on the EXAFS analysis in our previous manuscript. We also appreciate your careful inspection of the regularly spaced ripples for the MeCpPtMe₃ in **Figure 3b**. These ripples come from the too narrow width of the window “sill” dk (1.0 Å⁻¹ previously) of the Hanning window that could not minimize the truncation effect in the Fourier transform (FT). We also noticed that there are strong ripples below 1.4 Å, which do not bear any structural information but merely come from the low-frequency noise.

Therefore, we have made more efforts to reduce this background noise in the postedge background removal as well as to choose the more proper window “sill” width dk (3.0 Å⁻¹) of the Hanning window. The effect of dk on the FT curves is shown in **Fig. R10**. We can see that the dk of 3.0 Å⁻¹ can significantly suppress the ripples than dk of 1.0 Å⁻¹ do. While further increase dk to 4.0 Å⁻¹ would reduce the peak intensities.

Correspondingly, the Fourier transformed curves in Figure 3b-e are replaced by using Hanning window of dk = 3.0 Å⁻¹ in the revised manuscript. In addition, we added the details of XAFS data analysis and simulation in the “Methods” section, which is highlighted in yellow.

Fig. R10. Comparison of the Fourier transform curves for the MeCpPtMe₃ employing different widths of the Hanning window “sill” dk .

11) The reviewer would like to know the justification for using a k^3 weight for FT EXAFS data? Typically when EXAFS data is analyzed for lighter elements, a k^2 weight is employed. What is the justification for using a higher k weight?

Response: The reviewer is correct that k^2 -weight is commonly recommended for Fourier transforming the EXAFS data of the lighter element neighbors. Nevertheless, k^3 -weight has been also frequently utilized [Qiao, B., et al., *Nat. Chem.* 3, 634-641 (2011); Sun, S., et al., *Sci. Rep.* 3, 1775 (2013); Zhang, S. et al., *J. Am. Chem. Soc.* 135, 8283–8293 (2013)]. In fact, k^2 -weight or k^3 -weight has no physical difference in EXAFS data analysis.

Taking the Pt₂/graphene data as an example, we chose k^3 -weight rather than k^2 -weight, based on the following two considerations:

- In the k^2 -weighted $\chi(k)$ function, the high- k ($k > 8 \text{ \AA}^{-1}$) data is much lower in amplitude than the lower- k region data (Fig. R11a). When the $k^2\chi(k)$ function is Fourier transformed, it predominantly emphasizes the low- k region. Such a disadvantage could be compensated largely by k^3 -weighting (Fig. R11b), which strengthens the contributions of high- k region to the Fourier transform curve.
- Fourier transform of $k^3\chi(k)$ produces the peaks less broadening than k^2 -weighting. Hence the R-space peaks could be better-resolved, as indicated by the Fourier transform curves in the R-space (Fig. 11c, d).

Fig. R11. Comparison of the k^2 -weighted (a) and k^3 -weighted (b) $\chi(k)$ functions for the Pt_2 /graphene sample in the K -space. Comparison of the FT curves of the k^2 -weighted (c) and k^3 -weighted (d) $\chi(k)$ functions for the Pt_2 /graphene sample, under the otherwise identical FT conditions (k range of 2.2 – 13.5 \AA^{-1} , a Hanning window of $dk = 3.0 \text{ \AA}^{-1}$).

12) Supplementary figure 10 displays k^3 weight EXAFS oscillations and corresponding EXAFS simulations for different ALD experiments. Many of the simulation curves presented do not match up at all with the experimental. What is the reason for this? Furthermore, were Debye-Waller factors, Coordination number, or R-factors obtained from simulations? What are the best fit values from these simulations compared to your experimental data?

Response: As we have stated in the manuscript, due to the difficulty in discriminating the C/O neighbors by EXAFS fitting, we combined DFT calculations and EXAFS simulations to determine the optimized structures of the Pt_1 single atoms and Pt_2 dimers. The purpose of the simulations is to examine whether the simulated EXAFS data based on the DFT optimized structural models could reproduce the main features of the experimental data, therefore providing strong support to validate the DFT suggested structural models.

It is true that the simulated curves do not match well with the experimental data as the fitting curves do, because there is no any reiteration process to optimize the parameters such as interatomic distances, Debye-Waller factors, energy shift, and

others. During our simulations, the structural parameters such as coordination numbers and interatomic distances were set to the nominal values of the DFT optimized structural models. The amplitude reduction factor S_0^2 was fixed at the value of 0.86 which was determined by fitting the reference metal Pt foil. The Debye–Waller factors σ^2 for the nearest Pt–O/C and Pt–Pt pairs were set at the typical values of 0.0030 and 0.0065 Å², respectively, while σ^2 for all the other distant paths were set at 0.008 Å².

To further improve the match between the simulation and the experimental data for the MeCpPtMe/graphene sample, we optimized the two nearest Pt–C and Pt–O interatomic distances to 2.00 and 2.02 Å, respectively, both of which are **within ~2% error level** as compared to the optimized structure by DFT calculations (2.05 and 2.06 Å, respectively) shown in Supplementary Figure S11. **The optimized EXAFS simulation matches excellently with the experimental curve as shown in Fig. R12.**

While for Pt₁/graphene and Pt₂/graphene, the simulated EXAFS spectra based on the DFT-generated structures matched well with the experimental data (**Fig. 3d, e**), and therefore no further structure optimization was performed during EXAFS simulations.

All of these details are appended in the revised manuscript as we have mentioned. The updated simulations are included in the revised manuscript in Fig. 3. The changes are highlighted in yellow.

Fig. R12. Comparison of the EXAFS simulations based on the corresponding DFT calculated structural model (inset) with the experimental EXAFS spectra of MeCpPtMe/graphene. The ball in gray, white, red and dark blue represent carbon, hydrogen, oxygen and platinum, respectively.

13) There are number of grammatical and spelling errors littered throughout the paper. Ensure someone with an English background carefully looks through the paper before submitting again.

Response: We appreciate the reviewer’s kind suggestion. We have paid particular attention on the grammatical and spelling errors in the revised manuscript. **These changes are highlighted in red.**

Reviewer #3 (Remarks to the Author):

This paper describes the synthesis and testing of a series of catalysts made via the ALD of Pt precursors on defective graphite. The main achievement is the synthesis of catalysts that consist mainly of Pt dimers, which are argued to be very efficient in the dehydrogenation of ammonia borane. The synthesis of catalysts with atomic dispersions between the single atom and nanoparticle regime remains challenging so this is where the novelty of this paper lies. However, I have some fairly serious concerns about the experiments and data interpretation that should be addressed before publication. Furthermore, this paper needs a thorough edit by a native English speaker. The grammar and flow make reading the manuscript and SI very difficult at present. To name just a few examples of these problems – even the abstract is riddled with errors:

“Therein, one atom variation might....” This doesn’t make sense

“Progresses are made in the...” grammar

“Remains a grant challenge” I assume you mean grand?

“Pt single atoms deposition” should be “atom”

“45- folds...” should read “45-fold”

Note – these are just errors in the abstract – the whole paper and SI should be carefully edited and fixed.

Response: We appreciate the reviewer’s positive comments on our work, we also acknowledge the reviewer’s kind suggestion. We have paid particular attention on the grammatical and spelling errors in the revised manuscript. **These changes are highlighted in red.**

Major Scientific Points:

The biggest issue with the paper is that it is a report of the synthesis of Pt₂ catalysts and results of testing for dehydrogenation activity. The paper lacks clear explanations for:

1) The nature of the binding site on graphite – its speculative at present.

Response: We appreciate the reviewer’s valuable comment.

To reveal the nature of the binding site on the reduced graphene oxide support, we had done the following work.

(a) Determine the type of oxygen groups on the reduced graphene oxide support by systematic XPS measurements (**Fig. R8**). Phenol-related oxygen was found to be the dominant one on the reduced graphene oxide.

(b) Establish the relations between the content of phenol-related oxygen and the Pt loadings by a combination of XPS and ICP-AES measurements. We found that nucleation of Pt on graphene is directly related with that oxygen; without any phenol-related oxygen on the graphene outer surface, Pt does not nucleate on the graphene support at 250 °C, although there are a large amount of defects or graphene edges revealed by Raman spectroscopy in **Fig. R1b**. **Please see our detailed response to the comment #1 by the Reviewer 1#, and the comment #7 by the Reviewer 2#.**

Taken together, the combination of XPS and ICP-AES measurements strongly confirmed that the Pt is anchored onto the phenol-related oxygen, rather than on the graphene defect sites.

2) The exact nature of the Pt₂ complexes i.e. is C involved

Response: Based on the above discussion in comment 1#, we found that Pt does not anchor on the graphene defect sites. According to XPS measurements (**Fig. R8**), Phenols or phenol-carbonyl pairs [Bagri, A. *et al.*, *Nat. Chem.* 2, 581-587 (2010)] are clearly present on reduced graphene oxide to make the Pt deposition successful. Therefore, we could draw two conclusions as follows:

- (a) The two configurations with either one or two interfacial O atoms are both possible for the Pt₁ single-atom catalyst.
- (b) Directly bonding with two C atoms is not possible for the Pt₁ single atom.

Further analysis by combined DFT calculations and EXAFS simulations showed that the Pt₁ atom with one O and one C atom at the interface gave split FT peaks in the first shell, which is in contrast with the experimental results (**Supplementary Figure 13**). Therefore, the Pt₁ atom with two O atoms at the interface could be the more promising structure. Indeed, the EXAFS simulation based on this structural model optimized by DFT agreed very well with the experimental data (**Fig. 3d**), further validating this structural model.

Pt₂ dimers were “bottom-up” constructed from the Pt₁ single atoms, thus it is reasonable to expect that the interfacial atoms are also two O atoms for Pt₂ dimers, similar to that in Pt₁ single atoms. **Nonetheless, Pt₁ atom and Pt₂ dimers with one O and one C atom at the interface still cannot be completely ruled out. We pointed it out in our revised manuscript on page 10, highlighted in yellow.**

3) What about Pt₂ vs. Pt₁ makes these Pt₂ clusters active?

Response: This is an excellent question. We followed this reviewer’s constructive suggestions in the following comment #5, and performed DFT calculations to further illustrate the higher activity of Pt₂ dimers than that of Pt₁ single atoms.

In brief, we found that the relative lower adsorption energies of both AB and H₂ on Pt₂ dimers induced by the higher energy position of unoccupied 5*d* states above Fermi level, are likely the major reasons for the significantly higher activity than Pt₁ single atoms. **Please see our detailed response to the following comment #5.**

4) What is making some of the catalysts deactivate?

Response: We appreciate the reviewer’s valuable comment.

Recently, Chen *et al.* reported that B poisoning and sintering of Pt nanoparticles are the two major factors for the deactivation of Pt/CNT catalysts in the AB hydrolysis reaction [Chen, W. Y., *et al.*, *J. Am. Chem. Soc.* 136, 16736-16739 (2014)]. Bearing this in mind, we further evaluated the recycling stability of the Pt catalysts. We found that Pt₁/graphene, 2cPt/graphene, Pt/graphene-WI, Pt/carbon, Pt/SiO₂, and PtO₂ all deactivated considerably to different extent in the recycling test (**Fig. R13**). **Note that**

such catalyst deactivations are largely different from the dimeric Pt₂/graphene sample, where no visible deactivation was observed after 5 recycles (Fig. 6a).

Fig. R13. Recycling stabilities of various Pt catalysts in hydrolytic dehydrogenation of AB at room temperature for four recycles by adding additional 0.325 mmol of pure AB into the reaction flask after each run. (a) Pt₁/graphene; (b) 2cPt/graphene; (c) Pt/graphene-WI; (d) Pt/carbon; (e) Pt/SiO₂ and (f) PtO₂.

STEM measurements of these used catalysts revealed that Pt single atoms and nanoparticles all aggregated to different extents as shown in Fig. R14. Meanwhile, we also noticed that the Pt leaching on Pt/graphene-WI and Pt/SiO₂ during reaction was also considerable (Fig. R14c, e). Moreover, ICP-AES measurements of the used catalysts revealed that the adsorption of B species on the above samples were significant (Fig. R15). The moles of accumulated B to the moles of Pt surface atoms were even larger than one for all the Pt catalysts except Pt₂/graphene. Therefore, B poisoning of the above Pt catalysts is certainly possible in the AB hydrolysis reaction. In brief, aggregation of Pt single atoms and nanoparticles as well as B poisoning are the major reasons for the catalyst deactivation, consistent very well with the previous work [Chen, W. Y., et al., *J. Am. Chem. Soc.* 136, 16736-16739 (2014)].

The above discussion is added into the revised manuscript on page 12, highlighted in yellow. Figs. R13-R15 are also added in the Supplementary Information, as Supplementary Figures 22-24, respectively.

Fig. R14. The morphologies and Pt particle size distributions of the used Pt catalysts. (a) Pt₁/graphene; (b) 2cPt/graphene; (c) Pt/graphene-WI; (d) Pt/carbon; (e) Pt/SiO₂ and (f) PtO₂.

Fig. R15. The mole ratio of B element to Pt surface atoms in all used samples after four recycling test determined by ICP-AES.

5) The authors have performed DFT modeling of the Pt₂ clusters but don't present any electronic information about this cluster and how it might facilitate catalysis. This s, p,d state density information can be easily pulled from those calculations and should be compared to Pt₁ and Pt nanoparticles so to establish what is unique about the Pt₂ clusters. Furthermore, DFT should be used to calculate molecular binding strengths of the molecule of interest to these structures i.e. Pt₁ vs. Pt₂ vs Pt NPs and the reaction barriers for breaking B-H, N-H bonds and the barrier to H₂ formation quantified. The authors make some hand waving arguments based on the Sabatier principle to explain their findings but these additional DFT calculations would go much further to explaining why their Pt₂ systems works so well.

Response: We greatly appreciate the reviewer's constructive suggestions.

Following these suggestions, DFT calculations were further carried out to get a deeper insight into the vast activity difference between Pt₁/graphene and Pt₂/graphene.

AB is known as an excellent reducing agent [Andrews, G. C. & Crawford, T. C. *Tetrahedron Lett.* 21, 693-696 (1980)], and could likely stripe off the terminal dioxygen of Pt₁/graphene and Pt₂/graphene (the insets of Fig. 3d, e) during the reaction, thus partially reduced structures without the dioxygen were considered for both Pt₁/graphene and Pt₂/graphene (the insets of Fig. R16a). The reduced samples are denoted as Pt₁/graphene-R and Pt₂/graphene-R, respectively. It was found that the unoccupied 5d states of the top Pt atom in Pt₂/graphene-R locates at a considerably higher energy of 0.87 eV above Fermi level (E_f) than that of the Pt atom in Pt₁/graphene-R (0.40 eV), which indicates that Pt₁/graphene-R is more prone to accept electrons than Pt₂/graphene-R (Fig. R16a).

Fig. R16. (a) The local partial density of state (LPDOS) projected on the Pt 5d orbitals of Pt₁/graphene-R and the top Pt atom in Pt₂/graphene-R. The local configurations for AB adsorption on Pt₁/graphene-R (b), Pt₂/graphene-R(c). The ball in gray, white, pink, blue, red and dark blue represent carbon, hydrogen, boron, nitrogen, oxygen and platinum, respectively.

When AB is adsorbed on Pt₁/graphene-R, two B-H bonds were significantly elongated from 1.22 to 1.42 Å, with a strong adsorption energy of -3.20 eV (Fig. R16b). On Pt₂/graphene-R, the elongation of the two B-H bonds was slightly less, to 1.39 Å, and the AB adsorption energy was considerably weaker, about -2.81 eV (Fig. R16c). The adsorption of AB on Pt (111) was also investigated as a comparison (Fig. R17). We found that AB could quickly dissociate to three H atoms and BNH₃ species without any barrier. The AB adsorption energy is -3.97 eV, significantly stronger than that on Pt₁ single atom and Pt₂ dimer. The strong AB adsorption on Pt (111) revealed by DFT calculations agrees well with the literature, where Pt NP catalyst deactivation induced by B poisoning was observed during the AB hydrolysis reaction [Chen, W. Y., et al., *J. Am. Chem. Soc.* 136, 16736-16739 (2014)].

Catalyst deactivations along with considerable amounts of B adsorption were also observed on all the evaluated Pt samples, except the Pt₂/graphene (Figs. R13 and R15). Therefore, weakening the adsorption of AB on Pt appears to be an effective way to reduce B poisoning and to improve the catalytic activity. **Obviously, the considerable weaker adsorption of AB on Pt₂ dimer could be one key factor for its high activity as shown in Fig. 4.**

Fig. R17 The optimized adsorption configuration of NH₃BH₃ on Pt (111) surface from the top (a) and side views (b). The ball in white, pink, blue and dark blue represent hydrogen, boron, nitrogen and platinum, respectively.

H₂ adsorptions on Pt₁/graphene-R and Pt₂/graphene-R were also investigated as a descriptor of hydrogen desorption from the catalyst surface during the AB hydrolysis reaction (Fig. R18). Again, H₂ adsorption on Pt₁/graphene-R (-2.42 eV) is remarkably stronger than that on Pt₂/graphene-R (-1.29 eV). More interestingly, we found that H₂ chemisorbs dissociatively on Pt₁/graphene-R, but molecularly on Pt₂/graphene-R, indicated by the H-H bond distance of 2.02 and 0.96 Å, respectively. Such molecular adsorption of H₂ on the Pt₂ dimer with a moderate adsorption energy would favor the H₂ desorption during the AB hydrolysis reaction, thereby further boosting the catalytic activity.

In brief, compared to Pt₁ single atom, the higher energy position of the unoccupied state of the Pt 5d orbital of the top Pt in the Pt₂ dimer might play an important role in weakening the adsorption of both AB and H₂ molecules, which in turn facilitates the activity significantly.

The above discussion is added in our revised manuscript on pages 11-13, highlighted in yellow. Figs. R16 and R18 are combined together and added in the text as Fig. 5. Fig. R17 is also added in the Supplementary information as Supplementary Figure 21.

Fig. R18 The local configurations for H₂ adsorption on Pt₁/graphene-R (a), Pt₂/graphene-R (b). The ball in gray, white, red and dark blue represent carbon, hydrogen, oxygen and platinum, respectively.

6) The main point of the paper is that formation of Pt₂ clusters is self-limiting and on page 7 the authors claim that the reason for this is steric. This should be expanded upon as it is only one of several possible explanations that needs stronger support if it is to be used as the sole explanation of Pt₂ formation mechanism which is central to the paper.

Response: To “bottom-up” synthesize Pt₂ dimers on a high-surface-area support using ALD from gas phase, the following critical requirements have to be met:

(a) The reaction of Pt precursor with the support has to be self-limiting at the molecular level without any uncontrolled CVD growth or physical condensation. The self-limiting nature of ALD makes it possible in the ALD temperature window.

(b) The anchored Pt₁ single atoms should be stable under ALD conditions. This is very important to avoid any metal clusters/nanoparticles formation, so that the Pt₁ single atoms could become the only anchor sites for the second ALD cycle.

(c) Selective deposition of secondary Pt atom on the preliminary one. This is vital to avoid forming a mixture of Pt₁ single atoms and Pt₂ dimers.

To meet this critical requirement, precise control of surface nucleation sites is very necessary. It is well-known that all the ALD processes cannot grow on the “defect-free” graphene surface, because there is no any nucleation sites for anchoring metal ALD precursors. Very importantly, we noticed that Pt ALD does not nucleate on the graphene defect sites at a lower temperature of 250 °C either (Please see our detailed response to the comment #5, raised by the Reviewer #2).

Therefore, graphene turns out to be a very good candidate to create nucleation sites precisely through oxygen-functionalization using the well-established procedure of acid oxidation and thermal reduction [Ganguly, A., et al., *J. Phys. Chem. C* 115, 17009-17019 (2011); Yan, H., et al. *J. Am. Chem. Soc.* 137, 10484-10487 (2015)]. These oxygen functional groups would be the only nucleation sites on oxygen-functionalized graphene surface. They can be well controlled by changing the reduction temperature and time. Once Pt₁ single atoms are formed on the oxygen functional groups using saturated metal pulse times, these Pt₁ single atoms became the only type of nucleation sites for the secondary Pt ALD cycle, according to the self-limiting nature of ALD.

(d) Steric effect. The steric effect between chemisorbed metal precursors is a general phenomenon during ALD growth, please see the excellent review paper by Puurunen [Puurunen, R.L., *J. Appl. Phys.* 97, 121301 (2005)]. The steric effect as well as the self-limiting nature of ALD are both essential to make it successful to deposit one metal atom on one anchor site in one ALD cycle. Obviously, these anchor sites isolated from each other with a distance larger than the metal precursor size would be ideal for achieving this goal.

In this work, according to the STEM images in **Fig. 2b, c**, we found the distance between Pt₁ single atoms are greater than 2 nm in average,

significantly larger than the effective diameter of the MeCpPtMe₃ molecule of ~0.96 nm [Xue, Z.; Strouse, M. J.; Shuh, D. K.; Knobler, C. B.; Kaesz, H. D.; Hicks, R. F.; Williams, R. S. *J. Am. Chem. Soc.* **111**, 8779 (1989)]. This could be crucial to make all the Pt₁ single atoms accessible for chemisorbing the second MeCpPtMe₃ molecule in the second Pt ALD cycle without steric hindrance.

(e) Stability of the Pt₂ dimers under ALD conditions. Two successive cycles of Pt ALD on the reduced graphene at 250 °C resulted in the formation of a large amount of Pt NPs (denoted as 2cPt/graphene, shown in **Supplementary Figure 1**). In order to minimize the aggregation of Pt₂ dimer, the second Pt ALD cycle was performed at a significantly lower temperature of 150 °C.

In brief, synthesis of Pt₂ dimers on a high-surface-area support is very challenging. There are at least five critical requirements which have to be met. In the present work, both HAADF-STEM and ICP-AES provide direct evidence of the formation of Pt₂ dimers. It is worth noting that the rotation of Pt₂ dimers to the specific angles under the electron beam during STEM measurements provide strong evidence of the presence of Pt₂ dimers on graphene. Please see our detailed response to the comment #8, raised by the Reviewer #2.

Finally, the success in synthesis of Pt₂ dimers is also indicated by the observation of distinctly different catalytic performances in AB hydrolysis for Pt₁/graphene and Pt₂/graphene.

The above discussion has been emphasized through our revised manuscript on pages 4-8, as well as in the Conclusion Section on page 14, highlighted in yellow.

7) XAFS – the spectra of the Pt₁ vs Pt₂ in Fig 3 look identical to me and the simulation doesn't match the experimental data well at all. How can such strong conclusions be made from these XAFS spectra?

Response: We appreciate the reviewer's valuable comment.

EXAFS simulations alone could not draw the conclusion on the structures of the Pt₁ single atoms and Pt₂ dimers. The purpose of the simulations is to examine whether the EXAFS data based on the structural models optimized by DFT could reproduce the main features of the experimental data, therefore providing support to validate the DFT optimized structural models.

Excellent match between simulated curves and experimental data is not the purpose and could hardly be achieved by simulations, because there is no reiteration process to optimize the parameters such as interatomic distances, Debye-Waller factors, energy shift, and others as the curve-fittings do. **Please also see our detailed response to the comment #12 by the reviewer 2#.**

Nonetheless, we have to admit that other structural models still cannot be completely ruled out. For instance, Pt₁ single atom and Pt₂ dimers with one O and one C atom at the interface. We have pointed it out in our revised manuscript on page 10, highlighted in yellow. Please also see our detailed response to the comment #2 by this reviewer 3#.

8) Page 11 line 305 – I don't understand why changing the mole ratio of the reactant to the catalyst doesn't affect the rate? If you are studying the activity of catalysts you need to use a regime in which the intrinsic reaction rate dominates in order to make a comparison between catalysts. i.e. you double the catalyst amount you double the rate.

Response: On Page 11 line 305, we stated that “varying the mole ratio of Pt to the AB substrate did not significantly reduce the activity (Supplementary Figure 16)”, which is talking about the specific rates, not the hydrogenation generation rates (For convenience, Supplementary Figure 16 is shown here as **Fig. R19b**).

In order to avoid misunderstanding, we modified the sentence to “Pt₂/graphene could preserve the high specific rate to a large extent, when the mole ratio of Pt to the AB substrate was increased (Supplementary Figure 19)” in our revised manuscript on page 11, which is highlighted in yellow.

When the hydrogen generation rates are compared, we did find that double the catalyst amount will nearly double the hydrogenation rate (**Fig. R19a**). For instance, when the catalyst amount was 10 mg, it took about 55 sec to complete the reaction, but it took only about 33 sec for 20 mg catalyst. **The corresponding Supplementary Figure was updated using Fig. R19, highlighted in yellow.**

Fig. R19 (a) Plots of time versus volume of hydrogen generation by the dimeric Pt₂/graphene catalyst with different catalyst amounts. **(b)** The mass specific rates of Pt₂/graphene at the different ratios of n(Pt)/n(AB).

Minor Scientific Point:

Page 10 line 255-6 comparing Pt-Pt to Pt-O distances is not informative, the Pt-Pt distances for the Pt₂ complex should be given instead.

Response: We modified the text and stated that “our calculations show that the Pt-Pt bond distance in the Pt₂O₆ chain is 3.03 Å (the inset of Fig. 3e), consistent with the experimental results very well (Fig. 2g). The lengths of the Pt-O bonds in the Pt₂O₆ chain slightly vary from 1.93 to 2.03 Å.” in our revised manuscript on page 10, highlighted in yellow.

Reviewer #1 (Remarks to the Author):

The Most of questions were satisfactorily addressed in the revision. However, some issues need to be clarified.

1. The comparison of the catalytic activity was made based on the mole of Pt. It is fine to show how effective of the catalysts. However, it involved two effects of the Pt catalyst, namely Pt dispersion and site activity (TOF). The comparison in Figure 4 b should be based on TOF in stead of specific reaction rate to make it possible to show the superior site activity of the Pt dimers.
2. The last part in Page 19 is not very scientifically meaningful. Based on the TEM images, the Pt dimers sintered during the thermal treatment to form the nanoparticles. However, the TOF is significantly dependent on the particle size. The equation 4 is valid only the particle size is exactly identical to the one prepared by the impregnation. I suggest to remove this section.
3. The authors have added DFT calculation into the manuscript, providing a better insight for the effect of Pt catalysts (single and dimer) on the reactant and hydrogen adsorption and their reactions. The results seem to be in good agreement with DFT results in literature, where the top Pt in the dimer is negatively charged (Phys Chem Chem Phys, 16 (2014) 18586-18595), resulting in a lower adsorption of CO on dimer compared to single Pt on graphene (J Phys Chem C, 120 (2016) 12452-12462). It is better to cite the above two references.
4. In supplementary information Figure 6-9, it is not clear if it is the reduced sample.

Reviewer #2 (Remarks to the Author):

I read all replies from the authors. I think that the authors have addressed comments and suggestions well. I recommend that it is publishable now.

Reviewer #3 (Remarks to the Author):

Having re-reviewed this paper I am still in two minds. On one hand I think that the report of the synthesis of Pt₂ complexes and their potential utility is important as, after many reports of single

atom catalysts, two-atom heterogeneous catalysts are the next point of inquiry and haven't yet been realized to my knowledge. However, despite the revision much remains unclear about the nature of the Pt complex, the adsorption site on the support, the reaction mechanism etc. Rightly so the authors admit there are still things that are unclear. Therefore, I will leave this at the sum-average of the other reviewers' thoughts and the editor's discretion.

Reviewer #1 (Remarks to the Author):

The Most of questions were satisfactorily addressed in the revision. However, some issues need to be clarified.

1. The comparison of the catalytic activity was made based on the mole of Pt. It is fine to show how effective of the catalysts. However, it involved two effects of the Pt catalyst, namely Pt dispersion and site activity (TOF). The comparison in Figure 4 b should be based on TOF instead of specific reaction rate to make it possible to show the superior site activity of the Pt dimers.

Response: We appreciate the reviewer's valuable suggestion.

We calculated the TOFs based on the mole of surface Pt sites in the Pt catalysts. The results are shown in Fig. R1. Here the surface Pt sites were calculated using the equation (1)

$$Pt_{\text{surface}} = Pt_{\text{total}} \times d_{Pt} \quad (1)$$

Pt_{surface} is the number of surface Pt sites, Pt_{total} is the total number of Pt atoms, and d_{Pt} is the Pt dispersion which was calculated using the equation (2) according to the literature [R.D. Clayton, M. P. Harold, V. Balakotaiah, C. Z. Wan, Appl. Catal. B: Environ. 90 (2009) 662–676].

$$d_{Pt} = (1.1 / D_{Pt}) \times 100 \quad (2)$$

D_{Pt} is the Pt particle size.

In our revised manuscript, we included Fig. R1 into Supplementary Information as Supplementary Fig. 19 to keep in line with the literature where mass specific rates were previously reported in nearly all the studies for the AB hydrolysis reaction. The change on page 11 is highlighted in yellow.

Fig. R1. The TOFs of various Pt catalysts in the AB hydrolysis reaction based on the mole of surface Pt sites.

2. The last part in Page 19 is not very scientifically meaningful. Based on the TEM images, the Pt dimers sintered during the thermal treatment to form the nanoparticles. However, the TOF is significantly dependent on the particle size. The equation 4 is valid only the particle size is exactly identical to the one prepared by the impregnation. I suggest to remove this section.

Response: We removed the last part on Page 19 according to the reviewer's suggestion. Accordingly, the paragraph about the quantification of the survived Pt₂ dimers in the high-temperature treated samples on page 14 was also removed.

3. The authors have added DFT calculation into the manuscript, providing a better insight for the effect of Pt catalysts (single and dimer) on the reactant and hydrogen adsorption and their reactions. The results seem to be in good agreement with DFT results in literature, where the top Pt in the dimer is negatively charged (Phys Chem Chem Phys, 16 (2014) 18586-18595), resulting in a lower adsorption of CO on dimer compared to single Pt on graphene (J Phys Chem C, 120 (2016) 12452-12462). It is better to cite the above two references.

Response: We thank the reviewer for providing us the two excellent references. We added them in our revised manuscript on page 12 by stating that *"This result is in line with the recent literature where Åstrand et al. reported that Pt₁ single atom had a more strong ability to accept electrons than the top Pt atom in Pt₂ dimer, thereby showing stronger CO adsorption on Pt₁.^{61,62}"* which are highlighted in yellow.

4. In supplementary information Figure 6-9, it is not clear if it is the reduced sample.

Response: These are the as-prepared samples in the oxidized form, not the reduced samples.

In our revised manuscript, we further clarified it in the corresponding figure captions by stating that *"...the as-prepared Pt₂/graphene sample..."*. **The changes on pages S6-S9 in Supplementary Information are highlighted in yellow.**

Reviewer #2 (Remarks to the Author):

I read all replies from the authors. I think that the authors have addressed comments and suggestions well. I recommend that it is publishable now.

Response: We appreciate that the reviewer has satisfied with our previous response.

Reviewer #3 (Remarks to the Author):

Having re-reviewed this paper I am still in two minds. On one hand I think that the report of the synthesis of Pt₂ complexes and their potential utility is important as, after many reports of single atom catalysts, two-atom heterogeneous catalysts are the next point of inquiry and haven't yet been realized to my knowledge. However, despite the revision much remains unclear about the nature of the Pt complex, the adsorption site on the support, the reaction mechanism etc. Rightly so the authors admit there are still things that are unclear. Therefore, I will leave this at the sum-average of the other reviewers thoughts and the editor's discretion.

Response: We appreciate the reviewer pointed out that *"after many reports of single atom catalysts, two-atom heterogeneous catalysts are the next point of inquiry"*.

Reviewer #1 (Remarks to the Author):

The Authors have addressed all the comments well. I recommend publishing the paper.